# Automated Attention Pattern Discovery at Scale in Large Language Models

**Jonathan Katzy**                                    *J.B.Katzy@TUDelft.nl*
*Delft University of Technology*

**Razvan Popescu**                                    *R.M.Popescu@TUDelft.nl*
*Delft University of Technology*

**Erik Mekkes**                                       *E.Mekkes@Student.TUDelft.nl*
*Delft University of Technology*

**Arie van Deursen**                                  *Arie.vanDeursen@TUDelft.nl*
*Delft University of Technology*

**Maliheh Izadi**                                     *M.Izadi@TUDelft.nl*
*Delft University of Technology*

**Reviewed on OpenReview:** *https://openreview.net/forum?id=KpsUNOHAx7*

## Abstract

Large language models have found their success by scaling up their capabilities to work in general settings. The same can unfortunately not be said for their interpretability methods. The current trend in mechanistic interpretability is to provide precise explanations of specific behaviors in controlled settings. These often do not generalize well into other settings, or are too resource intensive for larger studies. In this work we propose to study repeated behaviors in large language models by mining completion scenarios in Java code datasets, through exploiting the structured nature of source code. We then collect the attention patterns generated in the attention heads to demonstrate that they are scalable signals for global interpretability of model components.

We show that vision models offer a promising direction for analyzing attention patterns at scale. To demonstrate this, we introduce the Attention Pattern – Masked Autoencoder (AP-MAE), a vision transformer-based model that efficiently reconstructs masked attention patterns. Experiments on StarCoder2 models (3B–15B) show that AP-MAE (i) reconstructs masked attention patterns with high accuracy, (ii) generalizes across unseen models with minimal degradation, (iii) reveals recurring patterns across a large number of inferences, (iv) predicts whether a generation will be correct without access to ground truth, with accuracies ranging from 55% to 70% depending on the task, and (v) enables targeted interventions that increase accuracy by 13.6% when applied selectively, but cause rapid collapse when applied excessively.

These results establish attention patterns as a scalable signal for interpretability and demonstrate that AP-MAE provides a transferable foundation for both analysis and intervention in large language models. Beyond its standalone value, AP-MAE can also serve as a selection procedure to guide more fine-grained mechanistic approaches toward the most relevant components. We release code and models to support future work in large-scale interpretability.

# 1 Introduction

The rapid adoption of large language models (LLMs) has intensified the demand for interpretability. Understanding how these models produce their outputs is essential for meeting regulatory standards, increasing user trust, and guiding performance improvements. Mechanistic interpretability offers a promising approach by tracing the internal flow of information within a model. This approach identifies circuits, combinations of neurons, attention heads, and residual pathways, that reveal how specific components contribute to a model's outputs and provide structural explanations of behavior.

Recent mechanistic interpretability methods, including sparse autoencoders (Cunningham et al., 2023), transcoders (Paulo et al., 2025; Dunefsky et al., 2024), and circuit-discovery techniques such as ACDC (Conmy et al., 2023) and path patching (Goldowsky-Dill et al., 2023), have revealed what features are encoded in LLMs and which circuits shape particular generations. Despite this progress, two major limitations restrict their broader adoption. First, the discovered circuits often fail to generalize across tasks, domains, or models (Lindsey et al., 2025). Second, constructing these circuits is computationally expensive, which limits their applicability in large-scale, real-world settings where data is noisy and less controlled (Lindsey et al., 2025; Anwar et al., 2024).

Our goal is to build on knowledge of transformer circuits and specialized attention heads to better understand how LLMs behave in large-scale, real-world settings. Here inputs are often noisy and less controlled than curated benchmarks. By doing so, we aim to generate actionable insights into which components of a model support accurate predictions and which may hinder performance. Understanding the contribution of individual components is the essence of fields such as circuit tracing. Being able to rapidly and cheaply identify critical components in an LLM allows us to apply more detailed but computationally expensive aforementioned methods in a targeted way, thereby enabling larger-scale investigations in the future.

To achieve these goals, we focus on an often-overlooked component of the transformer architecture: attention patterns. While most mechanistic interpretability methods trace how token representations evolve through the model, attention patterns reveal how information from the residual streams of different tokens is combined. This mixing of residual streams exhibits recognizable patterns that have been noticed in other works, however, they usually focus on one specific pattern (Sun et al., 2024), and identify patterns based on heuristics (Gopinath & Rodriguez, 2024). We want to replace the heuristics with a pattern mining approach for attention patterns.

Furthermore, to address the issue of generalizability beyond specific setups, we focus our work on code language models. Code has the unique property that we can mine common use cases using the abstract syntax tree. This lets us collect similar completion scenarios from a large dataset, rather than hand crafting these completions. More specifically, this investigation focuses on Java.

A central challenge is that existing interpretability techniques are designed for sparse, one-dimensional features, whereas attention patterns are inherently two-dimensional. This mismatch makes current tools poorly suited for analyzing them. To overcome this gap, we draw inspiration from vision models, which are specifically designed for structured two-dimensional data. In particular, we adapt the Vision Transformer Masked Autoencoder (ViT-MAE) (He et al., 2022) to reconstruct masked attention patterns. We refer to this model as the Attention Pattern Masked Autoencoder (AP-MAE).

We apply AP-MAE in three stages. First, we analyze the attention patterns it learns and connect them to existing findings in the literature (Section 5). Second, we use these patterns and their locations within an LLM to predict whether the model's generation will be correct, even without access to the ground truth, with an accuracy between 55% and 70% (Section 6). Finally, we leverage these insights to intervene on the LLM during generation by dynamically modifying its behavior to improve next-token accuracy by up to 13.6% (Section 7).

By introducing AP-MAE as a scalable method for analyzing attention patterns, we provide a cost-effective way to identify which components of a model warrant deeper mechanistic analysis. Our contributions are:

- We show that attention patterns can be effectively learned using vision-based models, and in particular demonstrate this with a masked autoencoder, establishing them as a tractable and informative object of study.

- We demonstrate that AP-MAE transfers across different models, with only a minor increase in loss when applied to models not seen during training.

- We establish that attention patterns alone can be used to classify the correctness of model predictions, in real-world settings, where information unrelated to the task may be present in the context.

- We show that dynamically removing heads identified in the classification task improves LLM performance for Java completion tasks.

- We find that performance collapses when too many of these heads are removed, showing that our method cost-effectively pinpoints the most critical heads to investigate with more detailed mechanistic approaches.

- We release our code[1] and pretrained models[2].

## 2 Related Works

**Mechanistic Interpretability of the Residual Stream**   The residual stream encodes the core representations that language models build during computation. Mechanistic interpretability seeks to uncover which features are stored in this stream and how they are transformed. Early work emphasized local structure, for example Sparse Autoencoders that map hidden activations to human-interpretable features (Bricken et al., 2023), probing methods that link residual states to the output head (Belrose et al., 2023), and transcoders that predict a module's output from its input (Paulo et al., 2025; Dunefsky et al., 2024). To extend beyond single locations, researchers have proposed attribution graphs (Dunefsky et al., 2024), crosscoders and replacement networks (Ameisen et al., 2025), and circuit-tracing approaches such as pruning and path patching (Conmy et al., 2023; Goldowsky-Dill et al., 2023). While these methods provide fine-grained insights, they are computationally expensive and often depend on carefully designed inputs, limiting their scalability and generalization (Lindsey et al., 2025; Anwar et al., 2024).

**Attention Heads as Functional Units**   Attention heads have emerged as particularly interpretable building blocks of transformers. Specific functional roles have been documented: induction heads support in-context learning (Olsson et al., 2022), iteration heads implement multi-step reasoning (Cabannes et al., 2024), and successor heads encode ordered successor functions (Gould et al., 2024). Other analyses reveal semantic concept vectors (Opiełka et al., 2025) and massive activations that dominate attention distributions (Sun et al., 2024). These roles are reflected in the distinctive attention patterns: diagonals for induction heads, vertical stripes for rare-token or letter heads, modular repetitions for successor heads, and dense blocks under activation outliers (Voita et al., 2019; Lieberum et al., 2023; Sun et al., 2024). Together, this body of work suggests that head-level functions and their associated attention patterns are two complementary perspectives on the same underlying circuits.

**Attention Patterns as Structured Signals**   Attention patterns provide a direct view of how tokens interact during computation, complementing approaches that focus on what a head computes. Prior work has documented recurring motifs, including induction diagonals (Elhage et al., 2021), rare-word vertical stripes (Voita et al., 2019), and anomalous patterns linked to errors (Yao et al., 2024). Surveys propose broader taxonomies of head roles (Zheng et al., 2024), while interventions exploit patterns for practical applications such as hallucination detection and mitigation (Chuang et al., 2024; Wang et al., 2025) or reasoning improvements via attention rebalancing (Li & Vargas, 2024). However, most prior analyses rely on manual inspection or heuristic rules (e.g., pattern visualizations or handcrafted thresholds), which do not scale and may overlook novel or subtle behaviors (Gopinath & Rodriguez, 2024).

---

[1] https://github.com/AISE-TUDelft/AP-MAE
[2] https://huggingface.co/collections/AISE-TUDelft/ap-mae

Table 1: Training and architecture parameters for AP-MAE.

| Model Inputs | | Encoder | | Decoder | |
|---|---|---|---|---|---|
| Pattern Size | 256 | Layers | 24 | Layers | 8 |
| Patch Size | 32 | Dim | 512 | Dim | 512 |
| Mask Ratio | 0.5 | Heads | 16 | Heads | 8 |
| Batch Size | 480 | MLP | 2048 | MLP | 2048 |
| Learning Rate | $1.44 \times 10^{-3}$ | | | | |

**Positioning**  These limitations motivate scalable methods that can automatically discover recurring attention patterns across tasks and models. By moving beyond heuristics and manual classification, such approaches could capture structured but previously hidden behaviors. Our work addresses this gap by introducing AP-MAE, which leverages vision-based architectures to efficiently learn, reconstruct, and mine attention patterns at scale.

## 3 Experimental Setting

A recurring criticism of mechanistic interpretability is that discovered features, attribution graphs, or circuits often depend on narrowly crafted inputs (Anwar et al., 2024). By this, we refer to task templates and manually designed prompts intended to trigger specific behaviors (Lindsey et al., 2025; Conmy et al., 2023), rather than experiments conducted on real-world data. Unlike these controlled inputs, real-world data is typically noisy and contains irrelevant context, meaning that curated prompts can strip away essential variability and risk producing insights that do not generalize beyond the benchmark setting.

To address this limitation, we use inputs that are not explicitly crafted for a task. We also build on findings that certain features are localized within a model, which motivates using a corpus that supports automatic extraction of comparable tasks. We focus on source code, as its structured syntax preserves the noise of real-world data while remaining analyzable. With parsing tools such as tree-sitter, we can mine consistent completion tasks at scale. We restrict our study to Java, whose verbosity provides abundant context and varied completion opportunities.

A key challenge with real-world data is contamination: overlap with model training corpora. While its impact on mechanistic interpretability remains unclear, we follow the standard practice of excluding training data. As most LLM training sets are undisclosed, we focus on the StarCoder2 (SC2) family (Lozhkov et al., 2024), which uniquely provides transparency about its corpus. To ensure decontamination, we base our experiments on The Heap (Katzy et al., 2025), a dataset deduplicated against SC2 training data.

## 4 AP-MAE

The core of our proposed approach is the Attention Pattern – Masked Auto-Encoder (AP-MAE) model for identifying patterns in LLM attention outputs. To train AP-MAE, we model the patterns as 1 channel images and base our model on the original ViT-L architecture (Dosovitskiy et al., 2021). We apply a novel scaling method to the attention patterns to allow the training to converge.

**Architecture**  AP-MAE is based on the ViT-L architecture with minor changes. We reduce the hidden size from 768 to 512 dimensions and scale the MLP down to 2048 parameters accordingly. We also remove the top right triangle of the data as this is masked in decoder-only attention patterns. For patches on the matrix's diagonal, we pad the masked values above the diagonal.

Finally, it has been shown that a small number of tokens receive the majority of the attention weight in any given attention head (Xiao et al., 2024). To prevent the reconstruction loss being dominated by the reconstruction of only the high attention tokens, we rescale all attention patterns using the natural logarithm. We choose a logarithmic function as it scales down large values more than smaller values, reducing the imbalance in intensities, and it is a monotonic (order preserving) function, meaning that any

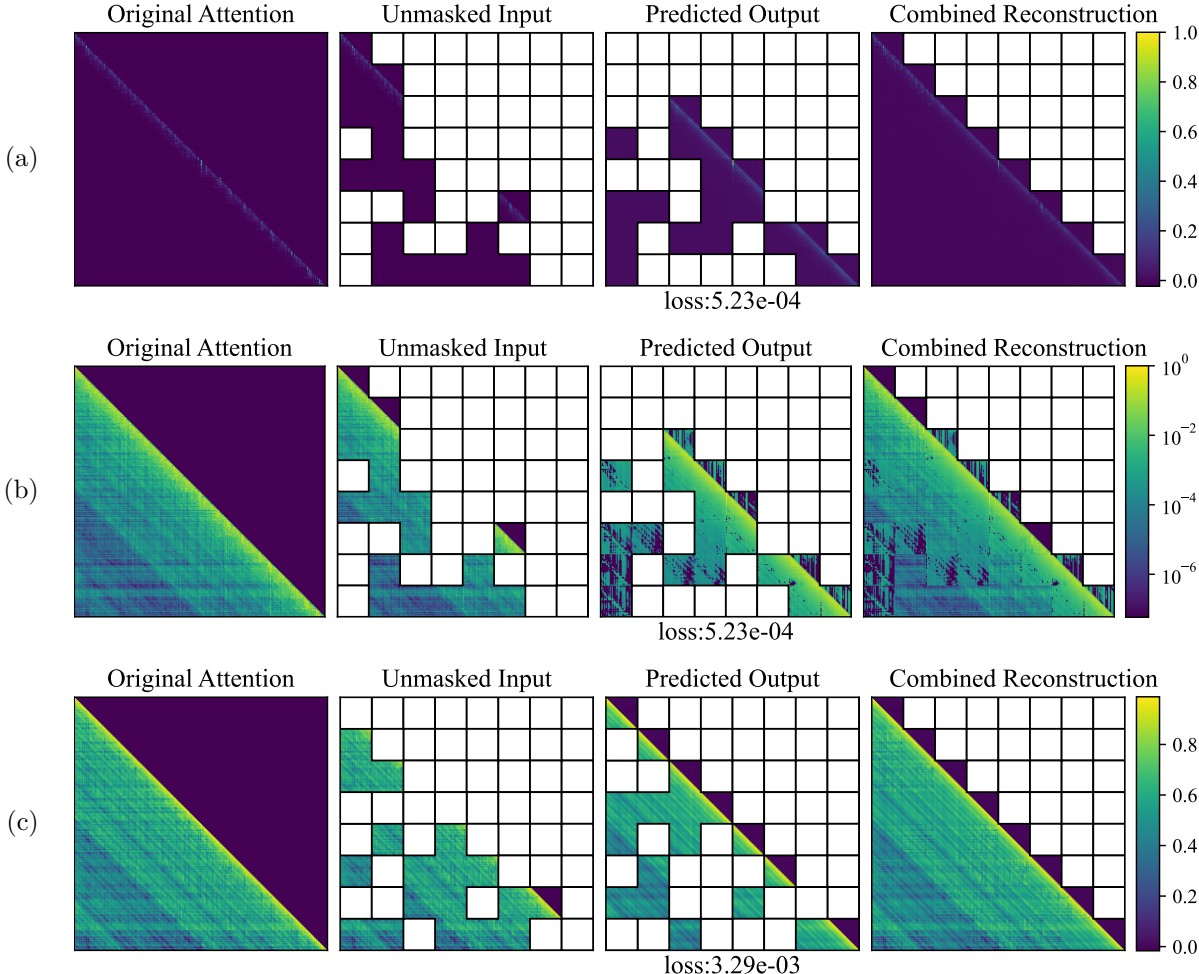

Figure 1: Comparison of attention pattern reconstruction methods: (a) raw attention pattern, (b) raw attention pattern with pixel values scaled for visualization, (c) log normalized attention pattern. Comparing the reconstructions in (b) and (c), we see that scaling the attention patterns prior to training AP-MAE is essential.

patterns based on the relative intensities will not change. This step is essential in allowing the model to fit to attention patterns, as demonstrated in the ablation study discussed below. Table 1 presents an overview of the architecture.

**Data**  To train AP-MAE we use attention patterns generated by SC2 3B, 7B, and 15B. To generate attention patterns, we mimic the training procedure used for training the SC2 models. We mask spans between 3-10 tokens in a code file, at random locations. We then add the same sentinel tokens to the input data as used during training. In order to ensure that the attention patterns used in this work are the same size, we truncate the context to exactly 256 tokens in total. As a final step in the training data selection, we focus exclusively on attention patterns generated when the prediction made by the model was correct, and we sub-sample the attention patterns from the language models.

Selecting for heads exclusively generated by correct predictions is a precaution to ensure the AP-MAE models will converge. From the logarithmic scaling study, we have seen that outliers in the attention scores can cause AP-MAE to not converge. It is currently unknown whether incorrect generations are the result of anomalous attention patterns, so we control for this factor. In future studies, AP-MAE could be used in an anomaly-detection setting to detect deviating attention patterns.

Table 2: Cross-evaluation of AP-MAE models. The Trained column refers to the target LLM that generated attention patterns the AP-MAE model was trained on. Evaluated refers to the target LLM that generated attention patterns that were used only for evaluation. Diagonal entries denote the traditional training/evaluation setup, while off-diagonal entries show the performance of AP-MAE on attention patterns generated by an 'unseen' model.

| Trained \ Evaluated | SC2 3B Loss ($\times 10^{-3}$) | SC2 7B Loss ($\times 10^{-3}$) | SC2 15B Loss ($\times 10^{-3}$) |
|---|---|---|---|
| SC2 3B | 7.07 (2.12) | 7.78 (2.05) | 9.53 (2.76) |
| SC2 7B | 7.55 (2.05) | 7.17 (2.12) | 9.29 (2.66) |
| SC2 15B | 9.57 (3.35) | 10.05 (3.79) | 7.59 (2.42) |

For each invocation of the target language model, we get 720 patterns for the 3B model, 1152 for the 7B model, and 1920 for the 15B model. As the generation procedure of these patterns is cheap, we sub-sample each generation to 25% of all attention heads. This allows us to get a larger variety of samples generated when the model is prompted in different locations in a code file. We ensure that when we subsample the patterns, we get 25% of the patterns from each layer. Other works have shown that there are signs of different behavior at different layers in a model, which we want to ensure we capture. In Figure 1 (c), we show an attention pattern on the left. Then we display the masking and the reconstructed parts of the masked pattern. Finally, we present the combination of the original and reconstruction.

**Training Setup** We train the models on eight A100 GPUs with a local batch size of 60 patterns. We train for $150,000$ batches using 72M attention patterns. The total training time is less than 100 GPU hours on our institution's cluster. Although we used eight A100 GPUs, all target models and AP-MAE models combined fit on one A100, making it possible to train on smaller servers. The limiting factor is the LLM size, as the AP-MAE encoder has 101M parameters and the decoder has 25M. We use the AdamW optimizer with a weight decay of 0.05 and a cosine annealing scheduler initialized with a global Learning Rate (LR) of $1.44 \times 10^{-3}$. We linearly upscale from the base LR of $1.5 \times 10^{-4}$ for a batch size of 50 used by ViT-MAE.

## 4.1 Generalizability

One of the main advantages of analyzing attention patterns, compared to representations of features within a model, is that they have the same dimensions between models. To investigate if AP-MAE can take advantage of this we evaluate its ability to reconstruct attention patterns from models it was not trained on. We cross-evaluate the AP-MAE models on a test set containing all combinations of SC2 target models. Table 2 provides an overview of the results. We observe that evaluating the encoder models on attention patterns from other target models results in a loss often within one standard deviation of each other. This shows that there are opportunities to use an AP-MAE base model to make inferences about a target LLM's behaviors without training a new model every time.

## 4.2 Ablation Study - **Logarithmic Scaling**

One of the preprocessing steps we used for the data is the log normalized scaling. To show that this step is necessary we conduct an ablation study by training an identical model, without scaling the attention patterns. In Figure 1 (a), we show an unscaled attention pattern together with its masking and reconstruction. Here it is difficult to see the reconstruction. In Figure 1 (b), we show the same pattern but scale the color gradient logarithmically to visualize the reconstruction. We see that the patches that were masked are corrupted. We compare this with the output of a model that has been trained on scaled attention patterns in Figure 1 (c). We see that when using logarithmic scaling, major patterns are reconstructed. While we cannot compare loss values directly, this shows the need for the logarithmic scaling of the attention patterns when training and evaluating the AP-MAE models.

## 5 Pattern Mining

### 5.1 Setup

We begin by encoding the attention heads using AP-MAE and select specific tasks to use as inputs to the language model, based on findings from previous research. We then cluster the resulting representations, a step that poses significant challenges given the large scale of the problem.

**Tasks**  Given the vast search space of possible circuits in Java, we leverage prior knowledge of circuits identified in LLMs to narrow our focus. Specifically, we select 11 tasks for pattern mining in attention heads, including a validation task in which the target model is probed with noise as an input.

1. Identifiers (1 task): One of the earliest benchmark circuits is the Indirect Object Identifier (IOI) circuit (Wang et al., 2023), where the model uses context to predict the correct token. Adapting this to source code, we mask a single identifier and task the target LLM with regenerating it.

2. Literals (3 tasks): Generating correct literals—spanning booleans, strings, and numbers—poses challenges distinct from identifier generation. Unlike identifiers, the correct literal values are not present in the input and must instead be inferred by the model (e.g., deducing $\pi = 3.14$). Prior work suggests that factual knowledge is encoded in the feed-forward layers of transformer models (Yao et al., 2022; Geva et al., 2020), making this setting particularly suitable for probing systematic behaviors of attention heads. Moreover, evidence of arithmetic-related circuits has been reported, including a greater-than circuit (Hanna et al., 2023) for numeric comparison and modules specialized for mathematical reasoning (Lindsey et al., 2025; Baeumel et al., 2025).

3. Operators (3 tasks): Beyond selecting appropriate operand values, recent work has shown that LLMs exhibit operator-specific heuristics when performing arithmetic reasoning (Nikankin et al., 2025). To complement literal selection, we include tasks focused on predicting the correct operator across three categories: boolean operators, arithmetic operators, and programming-specific assignment operators (e.g., +=).

4. Ending Statements (2 tasks): Finally, we evaluate the models' ability to complete complex syntactic structures. We consider two tasks. The first requires predicting the correct closing bracket for a statement, a capability that has been linked to specialized model circuitry (Ge et al., 2024). The second task involves predicting line endings. Unlike brackets, line termination in Java is not syntactically required, making this task a blend of program correctness and modeling human coding conventions. Prior work has examined structural and stylistic features at line endings in natural language (Lindsey et al., 2025).

5. Baselines (2 tasks): To contextualize our results, we consider two baseline tasks; random masking and noise. For random masking, we mask out a random span within a given code file. This is the same task that was used when generating the attention patterns to train AP-MAE. For the noise task, we create an input for the model of equal size to the other code tasks, however, instead of using a code file for the tokens, we sample tokens from the tokenizer of the target model using a uniform distribution over all tokens. This allows us to assess whether patterns formed by attention heads reflect meaningful structure, or whether they arise regardless of the model input. Patterns from this task are displayed in Figure 2(c).

**Encoding**  For encoding attention patterns, we employ our AP-MAE model, using the representation of the [CLS] token as the embedding, following standard practice in encoding pipelines. For each target LLM, we select $10,000$ input samples per task. These samples are balanced such that half correspond to attention heads associated with correct generations by the target LLM and the other half with incorrect generations. In contrast to the AP-MAE training phase, we do not perform head subsampling in this setting.

**Clustering**  Clustering all task samples is challenging due to both the high dimensionality of the representations (512 dimensions) and the sheer scale of the data (79.2M samples for SC2 3B, 126.7M for SC2 7B, and 211.2M for SC2 15B). The standard pipeline, dimensionality reduction with UMAP (McInnes et al., 2018) followed by clustering with HDBSCAN (Campello et al., 2013), is computationally infeasible in this setting, as it requires pairwise distance computations across the full dataset. Instead of resorting to subsampling, we decompose the problem into smaller, tractable subproblems. Specifically, for each model head, we first

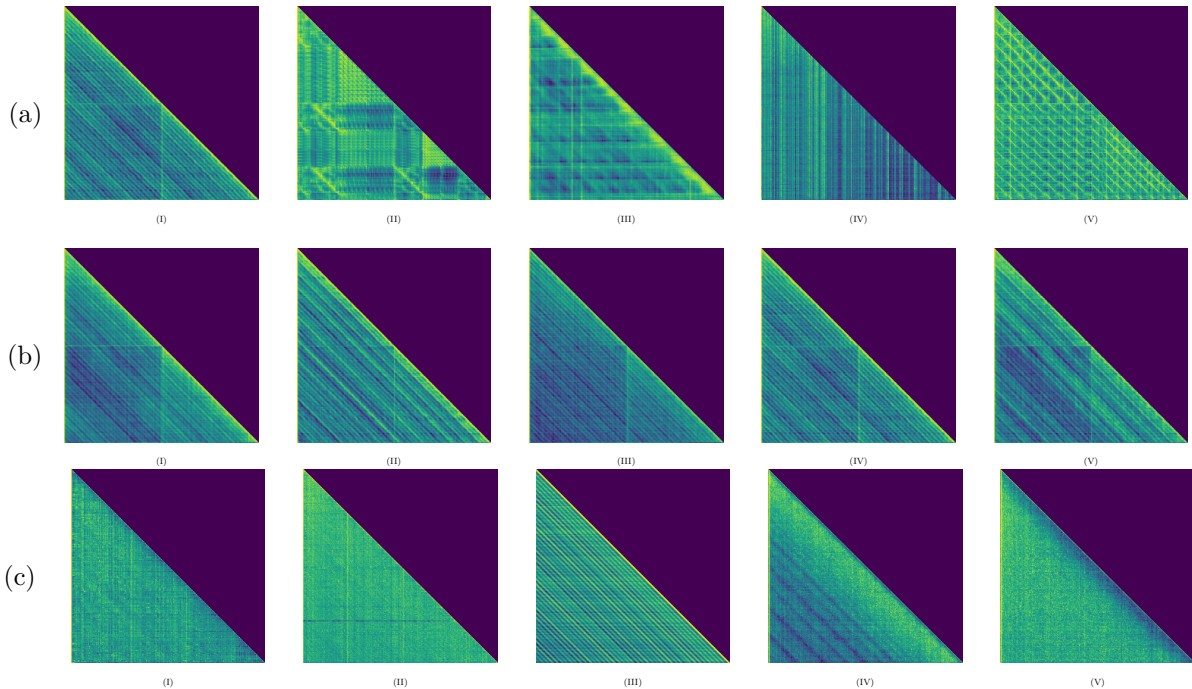

Figure 2: Comparison of different clustering results: (a) examples of different patterns found by clustering, (b) attention patterns within a single cluster, (c) attention patterns generated by noise, as described in Section 5.1.

reduce the representations to 8 dimensions with UMAP, and then cluster them with HDBSCAN. This yields up to 1920 independent clustering pipelines per model, each operating on approximately $110,000$ samples.

## 5.2 Identified Patterns

To assess whether repeated patterns exist and whether our clustering approach successfully captures them, we perform a qualitative evaluation of the resulting clusters.

Figure 2 presents three groups, each containing five attention patterns. Panel (a) illustrates five representative clusters, providing an overview of the variation observed across patterns. In pattern (a)(I), we observe a prominent diagonal structure: the highest attention scores concentrate around the preceding few tokens, with alternating bands of higher and lower scores extending outward. This behavior resembles an induction head, a specialized mechanism that facilitates in-context learning (Olsson et al., 2022). From patterns (a)(III) and (a)(IV), we observe that these heads feature high attention on individual tokens, as indicated by the vertical attention lines. This behavior has previously been characterized as LLMs allocating disproportionate attention to rare words in the input sequence (Voita et al., 2019) or individual letters (Lieberum et al., 2023). The high attention scores in (a)(III) around the diagonal in combination with the vertical lines, hint that some heads may exhibit multiple behaviors. In pattern (a)(III), we observe strong attention behavior reminiscent of the induction head, but distributed across multiple tokens. We also identify several recurring patterns that, to our knowledge, have not been previously documented. For instance, patterns (a)(II) and (a)(V) exhibit square-like structures with high attention diagonals that reappear at varying scales and frequencies across different heads; we highlight these two cases as representative extremes. This description would match the definition of global patterns given by (Gopinath & Rodriguez, 2024), however, the examples provided look distinctly different from the discovered patterns, showing that there is a need for a more detailed taxonomy of attention patterns.

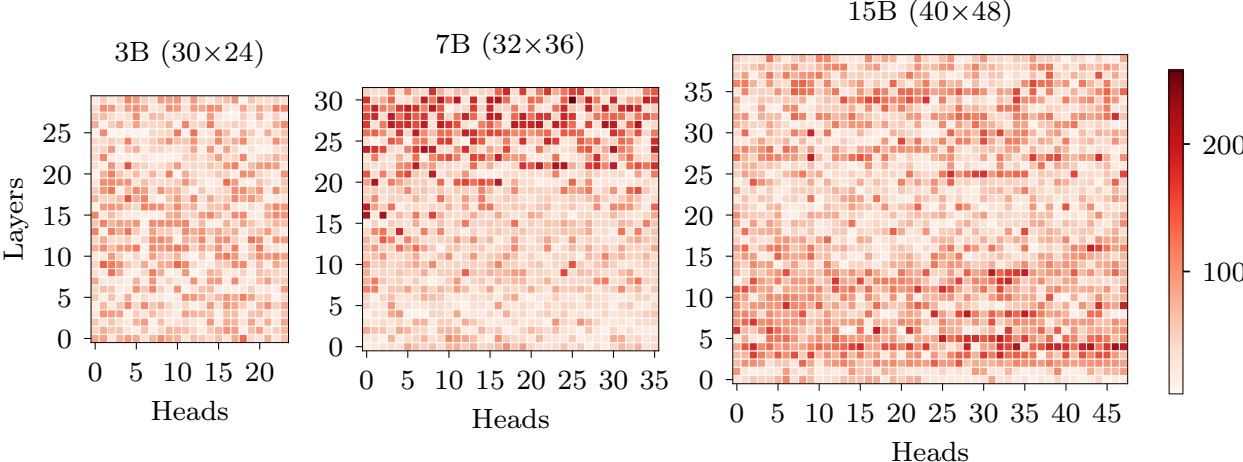

Figure 3: Distribution of the number of clusters in a head

In addition to capturing distinct patterns, our method demonstrates robustness to variations in the ordering of input tokens. Figure 2(b) illustrates five patterns grouped within the same cluster (Figure 2(a)(I)). Although the intensity and position of the global diagonal lines vary, the general pattern is preserved. AP-MAE can capture these differences in pattern locations, allowing it to handle changes in the ordering of input tokens. Finally, we investigate whether heads generate patterns regardless of the input. To this end, we visualize the heads (Figure 2c) generated when feeding the model random noise, as described in Section 5.1. Unlike Figures 2(a) and 2(b), only a few discernible structures emerge. The most recognizable case is (c)(III), which closely resembles patterns observed in Figure 2(b), and stands as the most similar pattern we were able to identify. Notably, the characteristic square structures seen in (a)(II) and (a)(V) do not appear under noisy inputs. This absence suggests that such square patterns may serve as strong indicators of heads engaging in meaningful computation, an observation that highlights a promising direction for future research.

Finally, we discuss the use of masking as a training objective and our choice of encoder. As discussed previously AP-MAE has been shown to generalize away from token positions by grouping the same global patterns (vertical/horizontal/diagonal lines) into separate clusters. However, we do see that the similar motifs in Figure 2(a)(II) and (V) are not in the same cluster. This is a limitation of MAEs as a whole, they focus on a complete image, rather than seeing images as compositions of basic patterns at different scales. The results from our large-scale study reveal the consistency of smaller patterns present within attention patterns. This opens the door to using more specialized models, such as CNNs, to locate specific motifs, such as the square structures, and investigate attention patterns as a composition of different motifs.

## 5.3 Pattern Distribution

We next examine how the discovered patterns are distributed across attention heads. Figure 3 reports the number of clusters identified in each head. A consistent trend emerges: as model size increases, a subset of heads produces a broader variety of patterns. In the 3B model, most heads yield only a few clusters, whereas the 7B model exhibits substantial diversity, particularly in later layers. The 15B model shows fewer heads with high diversity compared to the 7B model, though some still capture a wide range of patterns. These differences in distribution suggest increasing specialization of certain heads, potentially enabling them to handle a broader spectrum of noisy inputs.

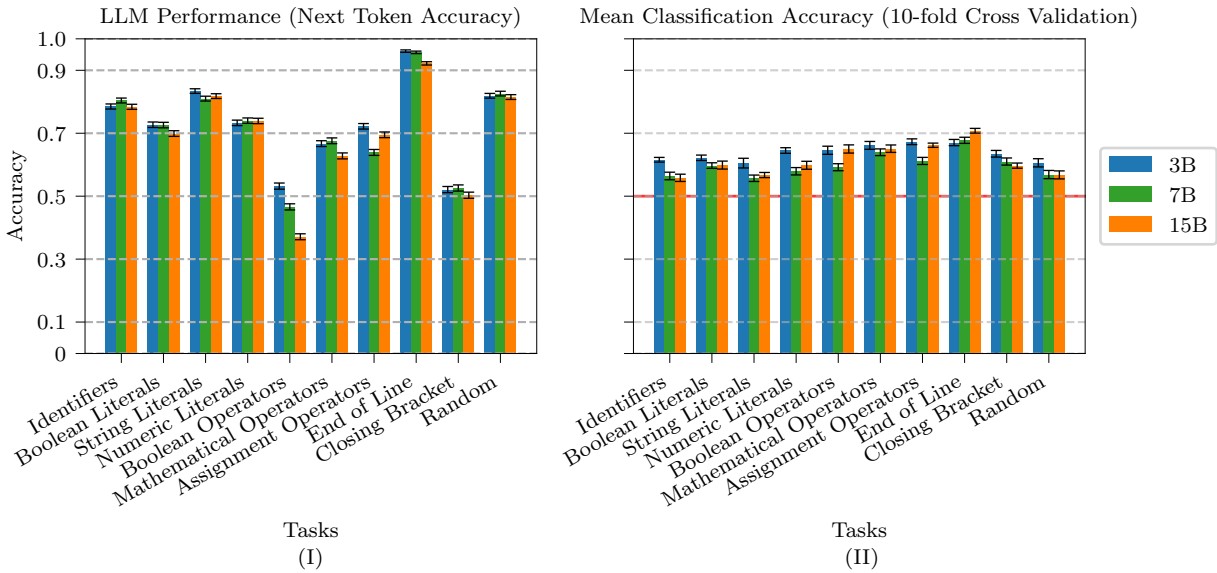

Figure 4: Performance of target LLMs on the studied tasks, and the accuracy of the CatBoost classifier

# 6 Classification

## 6.1 Setup

To determine whether a target model's prediction is correct, we treat the output of each head as a categorical feature, using the cluster assignment of that head for the given prediction as its value. This formulation yields a tractable prediction problem with between 720 features (SC2 3B) and 1920 features (SC2 15B). We perform classification at the task level, training a dedicated predictor for each task. The Noise task is excluded, as it does not admit a notion of correctness.

For classification, we employ a gradient boosting decision tree model, CatBoost (Prokhorenkova et al., 2018). CatBoost offers two key advantages for our setting: (i) it enables the computation of SHAP values (Lundberg & Lee, 2017), which we use to quantify the contribution of each feature (here, individual transformer heads) to the distinction between correct and incorrect predictions, and (ii) it natively handles categorical data, eliminating the need for additional preprocessing.

## 6.2 Performance

We give the results of the classification task in Figure 4 (II). We also include the performance of the target models in completing the next token prediction task in Figure 4 (I), to see if performance has an effect on our ability to correctly classify a prediction as correct or incorrect. For the classification task we plot the mean accuracy and the 95% confidence interval over a 10 fold cross validation using a 90, 10, 10 split. We see that the performance varies little between runs due to the small range of the 95% confidence interval. The first thing we see in Figure 4 is that there is no correlation between classification performance, and target model performance. We also investigated the drop in performance for boolean operators and closing brackets, and see that it is indeed due to the models themselves, explaining this is worthy of an investigation of itself. Next, focusing only on Figure 4(II), we see that some tasks are indeed harder to classify as correct than others. Tasks such as Identifier, Boolean Literals, and String Literals perform similarly to the Random masking task, with accuracy scores between 55% and 60%. The best performing task is predicting the End of Line token, which has a mean accuracy of just over 70% for the 15B model.

To investigate which parts of the LLMs are needed to differentiate between a correct and incorrect prediction, we use the maximum difference between mean SHAP values per category at each head. This highlights

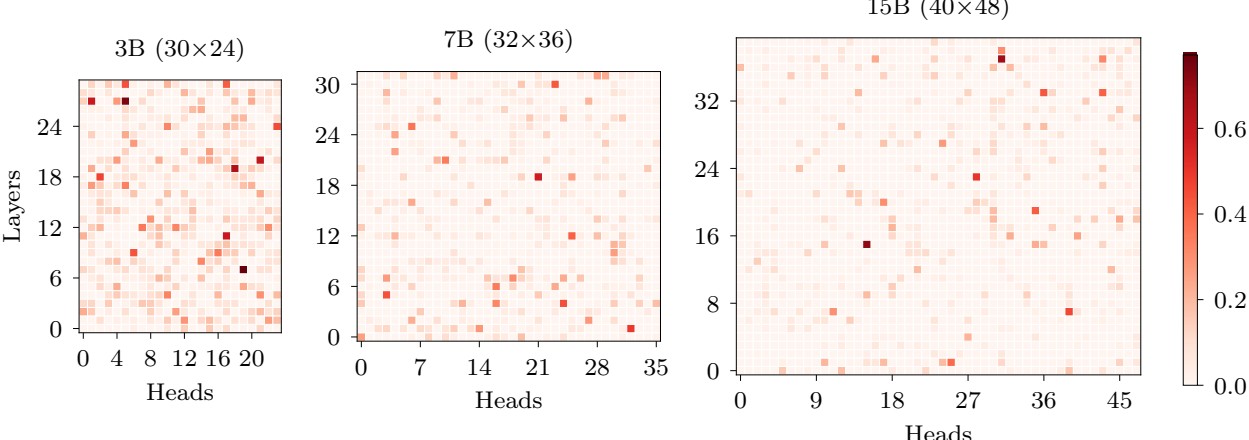

Figure 5: Difference in mean SHAP values per cluster for the CatBoost classifiers, classifying predictions for the End of Line task across all target sizes

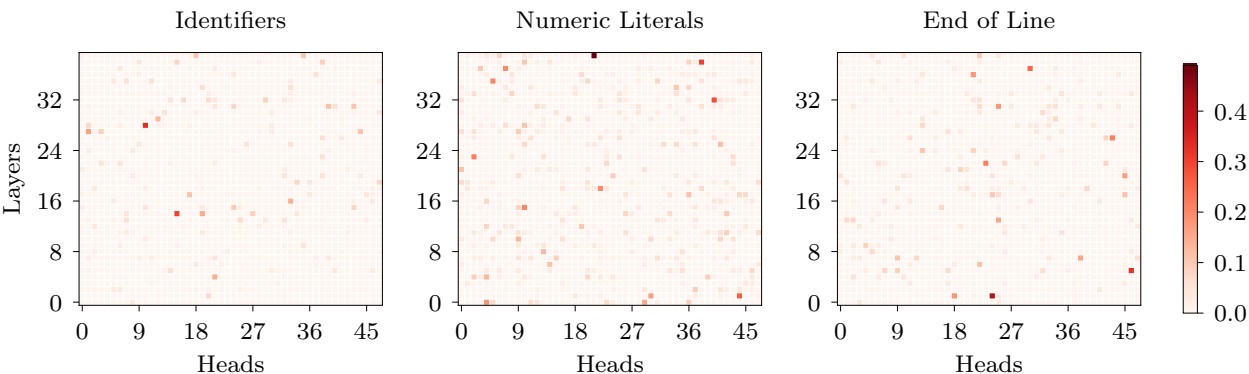

Figure 6: Maximum difference in mean SHAP values per cluster explaining the global effect of each pattern in each head on the correctness classifier

heads that are both strong indicators of being a correct or incorrect prediction, depending on the pattern we detected in them. We plot these values in Figure 5, the mean SHAP values of each head are listed in appendix A. Here we focus on the end of line token task. We see that the plots are sparse; a small number of heads is enough to differentiate between correct and incorrect predictions. Furthermore, sparsity increases with model size hinting that heads get more specialized as models increase in size. This allows us to highlight heads that are of interest when determining where LLMs make mistakes.

Finally, we investigate differences between tasks when it comes to predicting if an LLM generation is correct. To investigate this, we plot the same difference between mean SHAP values introduced earlier for different tasks targeting the 15B model in Figure 6. Here we see that for each task, the SHAP values are sparse, and different heads are highlighted. Showing that classification value of patterns is task dependent.

## 7 Intervention

### 7.1 Setup

Having established that attention patterns can distinguish correct from incorrect generations, we next intervene on the model by selectively setting the contribution of specific attention heads to zero, based on their importance for classification. Importance is quantified using SHAP values derived from the classifier model. For each task, we select heads with the largest SHAP values (positively associated with correct pre-

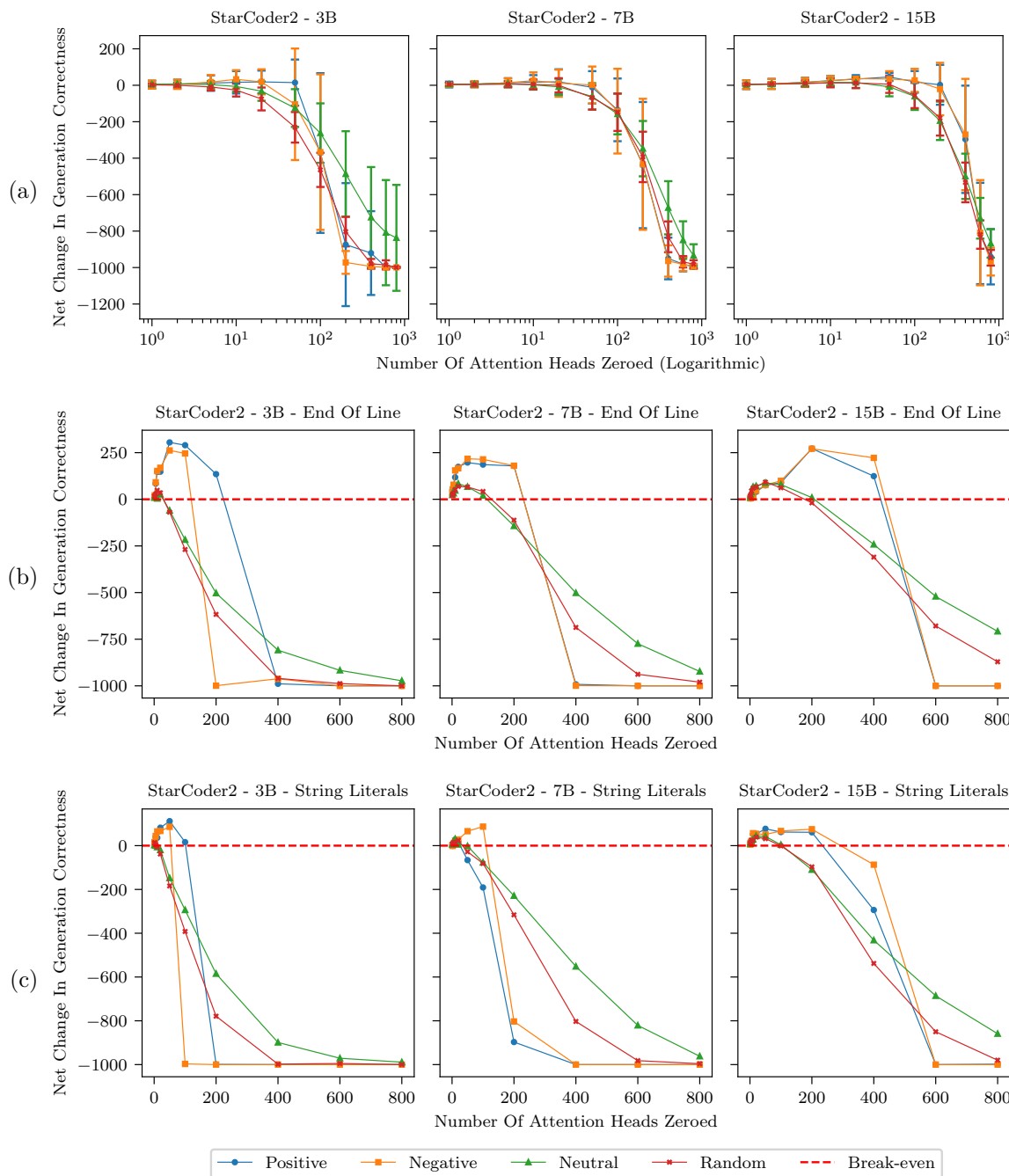

Figure 7: Effects of interventions based on SHAP values showing the net change in correct token generation as heads are progressively zeroed. (a) Global results across tasks, shown on a logarithmic x-axis for clarity, (b) End-of-Line token prediction, the best-performing classification task, (c) String Literal completion, the worst-performing classification task,

dictions), the smallest values (associated with incorrect predictions), and values near zero (neutral influence). From each group, we progressively zero out subsets of heads, ranging from 1 up to 800 where available. As a baseline, we also evaluate the effect of zeroing an equal number of randomly selected heads. For each condition, we run 1,000 inferences on generations the model originally predicted correctly and 1,000 on those it predicted incorrectly, recording whether the intervention changes the predicted token. For incorrect generations, we additionally record whether the new prediction matches the correct token.

### 7.2 Results

The effects of our interventions are summarized in Figure 7(a), which reports the net change in accuracy across tasks (originally incorrect predictions corrected minus originally correct predictions lost). Positive values indicate accuracy gains, negative values indicate losses. At the global level, a consistent pattern emerges: zeroing the contributions of a small subset of heads with extreme SHAP values yields accuracy improvements, but beyond a threshold, performance collapses rapidly. Unlike the gradual declines observed when randomly zeroing heads or removing near-neutral SHAP heads. This collapse occurs because all originally correct predictions change to incorrect, whereas none of the incorrect predictions convert to correct. The large error bars reflect task-level variability in the point of collapse and the size of initial gains.

To examine this behavior at the task level, we highlight two representative tasks in Figure 7(b) and (c), showing the best-performing classification task, End-of-Line token prediction, and the worst-performing task, String Literal completion. For the End-of-Line task, zeroing the contributions of a small number of heads with the most positive or most negative SHAP values increased the number of correct predictions by up to 271 for the 15B model. In contrast, the maximum improvement for the String Literal task was 112 predictions for the 3B model under the same conditions. The number of heads required to trigger a drastic drop in performance also differed between tasks. For the 3B model, collapse occurred after zeroing 200 positive and 100 negative SHAP heads in the String Literal task, and after zeroing 400 positive and 200 negative heads in the End-of-Line token task. For the 7B model, this threshold was approximately 200 heads for both positive and negative SHAP conditions in the String Literal task, increasing to 400 heads in the End-of-Line token task. These differences in both the magnitude of gains and the point of rapid decline help explain the high variance observed in the global results and underscore the critical role of task-specific circuits in model behavior.

Overall, these results demonstrate that while suppressing selected heads can improve accuracy in specific tasks, excessive removals change all correct predictions to incorrect without producing any new correct predictions. From prior work (Ameisen et al., 2025; Conmy et al., 2023), we have seen that only a small subset of the components in an LLM are necessary to generate a correct prediction, known as circuits. However, it has been postulated that there are multiple (secondary) circuits in a model that can also be used to generate a correct prediction (Rai et al., 2024). We believe that the performance gain observed when removing a small number of heads is due to a resource race within the model being resolved between competing circuits. The final collapse happens when the last remaining circuit has been severed and so the model can no longer complete the task. This does, however, require more research. Finally, the magnitude of this collapse threshold varies by task, highlighting the importance of task-specific circuits in maintaining prediction accuracy.

## 8 Limitations and future work

Despite the insights provided by our large-scale analysis of attention patterns in LLMs, several limitations remain. First, computational constraints required a fixed input length, which facilitated fair comparisons across samples but does not capture the full variability of sequence lengths. Future work should investigate how attention patterns evolve under varying input lengths, particularly with respect to encoding and clustering behaviors.

Second, we relied on a vision transformer architecture due to its scalability, robustness to noisy inputs, and ability to generalize under potential distribution shifts. These are valuable characteristics for an initial large-scale investigation into the consistency of attention patterns. However, they do come at the cost of needing to train a computationally expensive architecture. Having shown that attention heads generate consistent patterns, it creates the opportunity to use computationally cheaper models such as Bags of Visual Words (Gidaris et al., 2020), VLAD (Jégou et al., 2010) or CNNs (O'Shea & Nash, 2015) to detect specific patterns or in low latency settings. In this work we have not included these methods as baselines, as the focus is on discovering the variability of patterns, future work should focus on using these findings and improving the insights we can mine from such a data-centric approach to mechanistic interpretability.

Third, our analysis focused on a subset of downstream tasks, selected based on parsing rules and targeting known behaviors, such as in-context learning, factual recall, and established mechanistic circuits, however, there was a large amount of variance between performance from task to task. While this approach allowed us to capture representative behaviors, it may overlook more subtle or emergent patterns, such as distinctions between the first token of an identifier versus subsequent tokens, or other fine-grained sequence-level behaviors. Future studies could leverage online learning or large-scale behavior mining to systematically explore a broader range of tasks and data streams, supporting applications such as fault localization and transfer learning. Similarly, our selection of tasks is designed to be similar to other tasks studied in related works, however there is no formalization of what 'tasks' exist or how they can generalize across different settings. Formalizing the tasks that are of interest for mechanistic interpretability in LLMs as a whole would be an important step toward maturing the field of circuit discovery.

In combination with the formalization of tasks, this investigation focused exclusively on code (Java). This is due to the highly structured nature of code that lets us mine repeated patterns from large corpora of code files. In order to extend this work to natural language, a parser for natural languages is required (Grune & Jacobs, 2008). These do exist, however, natural languages are not as rigid in forcing correct syntax on users as programming languages. This leads to many ambiguities when parsing natural text and has led to the adoption of probabilistic parsers (Jelinek et al., 1992). Similar to our decision to train the AP-MAE models exclusively on correct predictions, it is unknown what effect incorrectly or inconsistently parsed files could have on the results of these experiments. Future studies should investigate the robustness of circuit tracing works when working with probabilistic parsers, however it is beyond the scope of this investigation.

Finally, our method is designed to identify patterns rather than provide explicit explanations. It efficiently highlights regions of interest in a model, but does not reveal detailed mechanisms. Combining this approach with fine-grained mechanistic interpretability methods presents an important avenue for future work, potentially enabling a deeper understanding of task-specific circuits and their role in model behavior.

## 9 Conclusion

We introduced AP-MAE, a vision-inspired approach for analyzing transformer attention at scale. AP-MAE reconstructs attention patterns with high fidelity, generalizes across model sizes, and predicts correctness without access to ground truth. Guided by SHAP values, targeted interventions improve model accuracy, while excessive removals prevent any correct generations.

These results establish attention patterns as a transferable signal for probing model behavior and position AP-MAE as a cost-effective complement to fine-grained mechanistic methods. By filtering attention heads worth deeper analysis, AP-MAE bridges behavioral evaluation and circuit-level discovery, making interpretability more scalable in real-world, noisy settings.

Looking forward, AP-MAE opens several avenues for advancing interpretability. Integrating it with mechanistic methods could reveal the causal mechanisms underlying discovered patterns. Expanding the analysis beyond code completion to diverse domains may uncover new classes of structured behaviors. Finally, developing adaptive interventions that modulate rather than disable heads could enable more robust performance gains while preserving critical computation. Together, these directions highlight AP-MAE as a practical foundation for scalable and generalizable interpretability.

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

# A   SHAP values per task

Due to the large number of plots, we include only a select few in the main body of the paper. In this section we include the plots of all average SHAP values for each task and model. These were the values used to select which heads to zero in the intervention stage. As can be seen the plots are very sparse, so often fewer than the upper limit of heads was zeroed out.

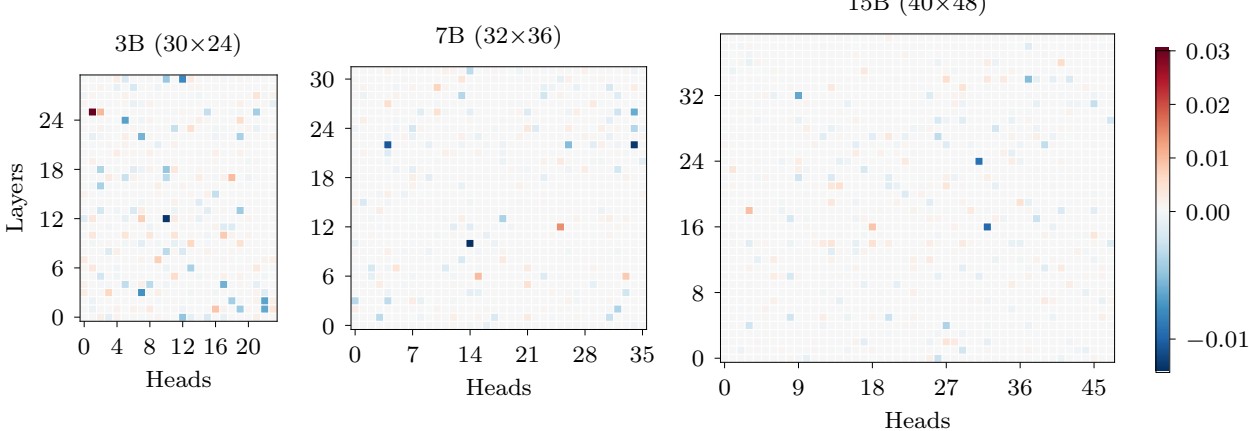

Figure 8: SHAP values for the assignment operators task.

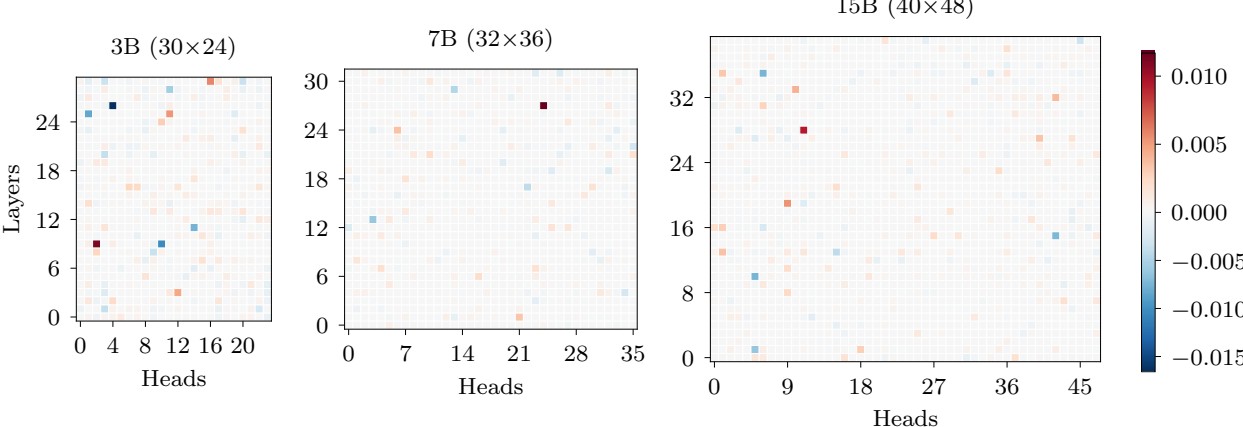

Figure 9: SHAP values for the boolean literals task.

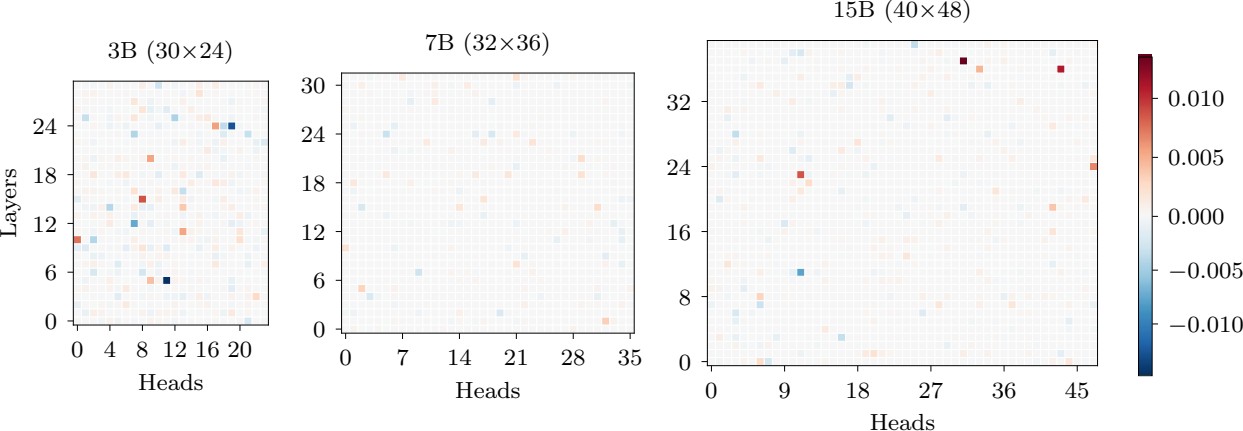

Figure 10: SHAP values for the boolean operators task.

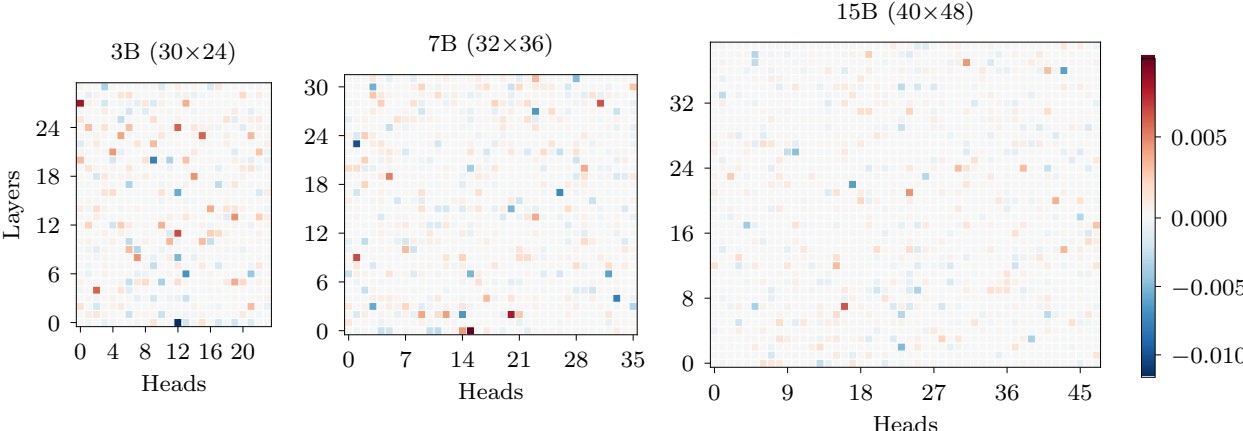

Figure 11: SHAP values for the closing brackets task.

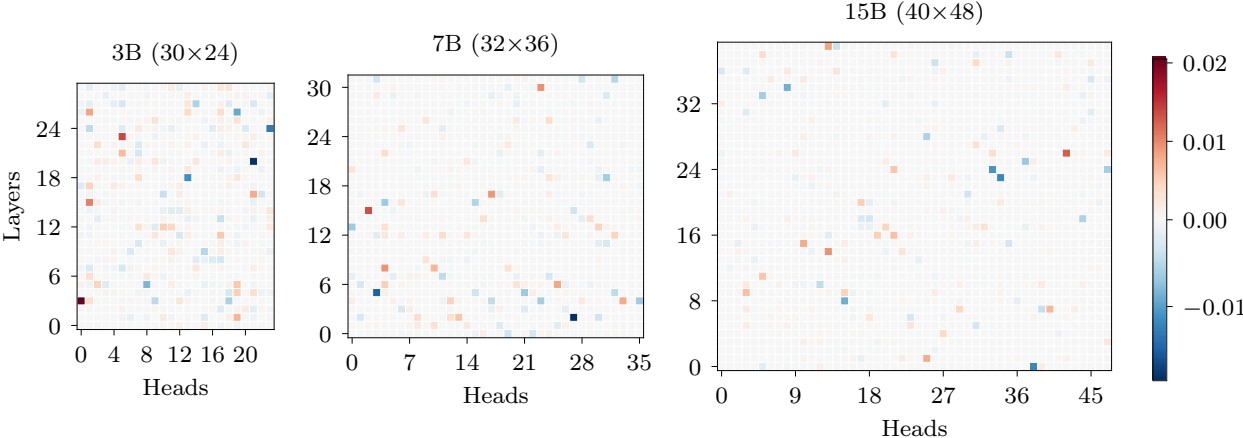

Figure 12: SHAP values for the end-of-line token task.

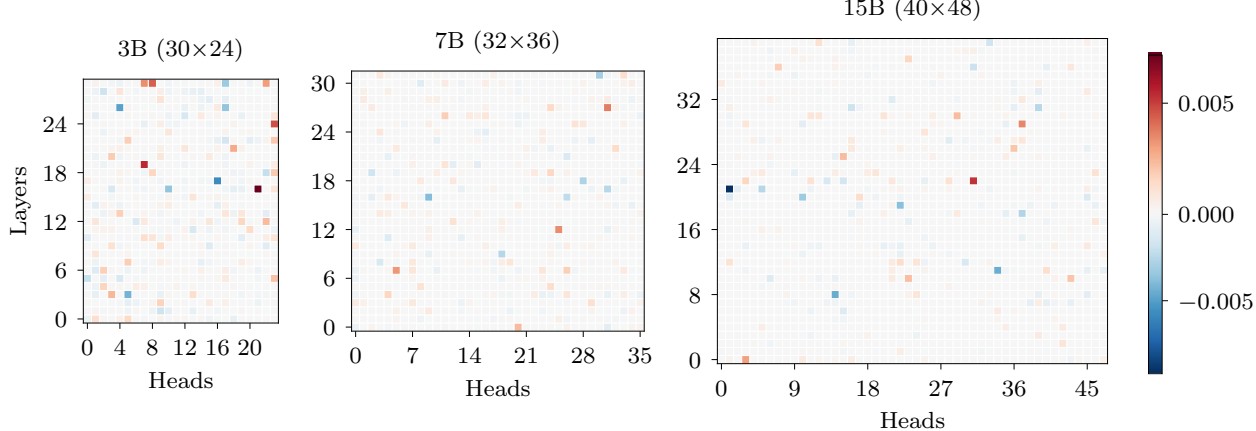

Figure 13: SHAP values for the identifiers task.

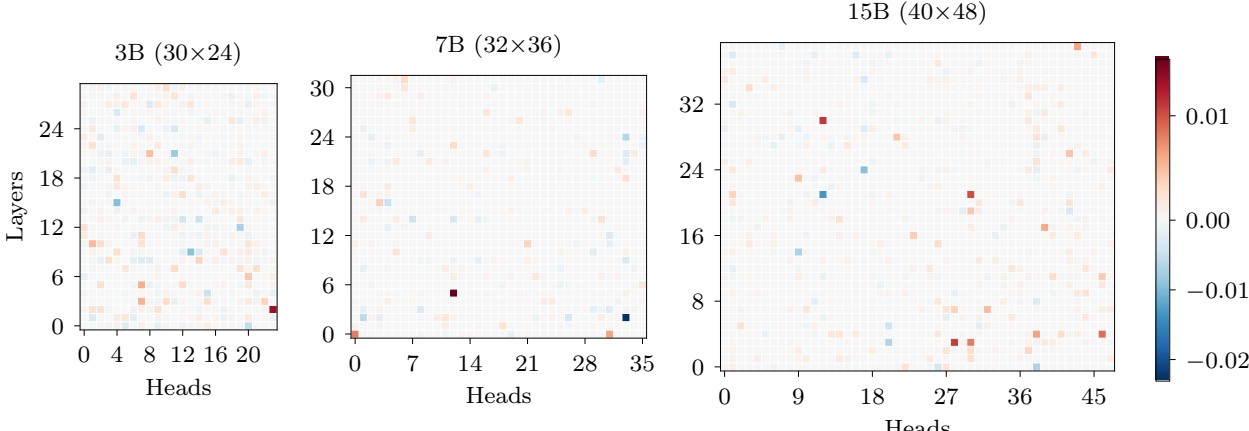

Figure 14: SHAP values for the mathematical operators task.

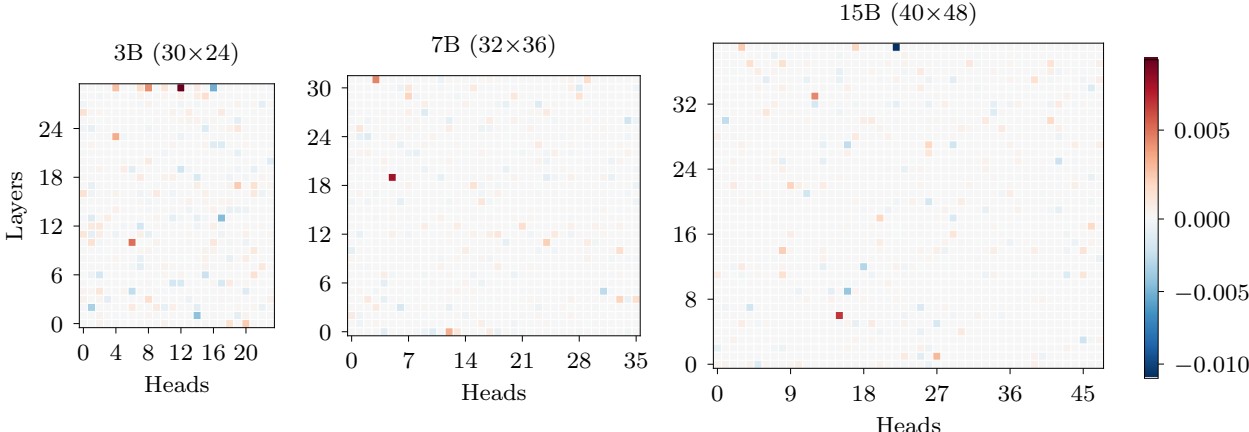

Figure 15: SHAP values for the numeric literals task.

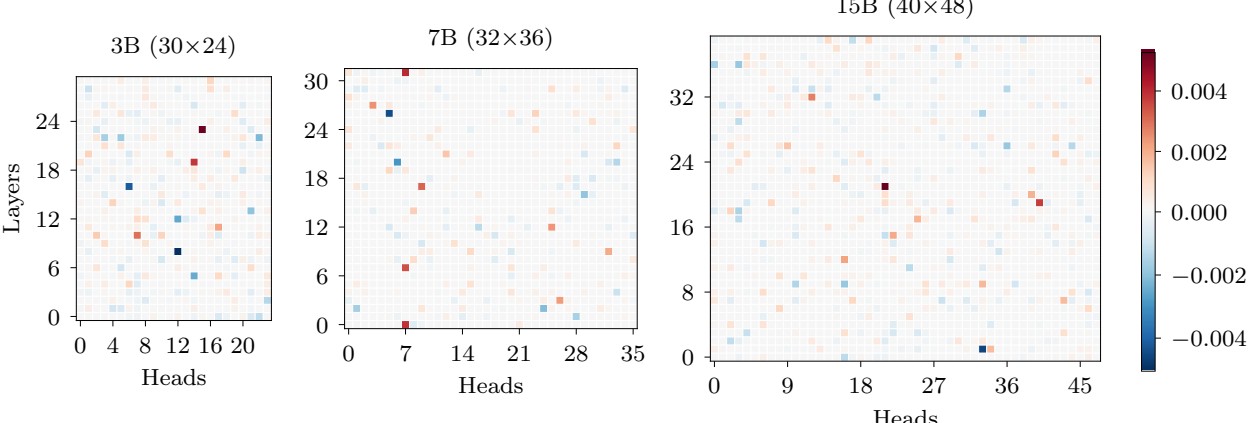

Figure 16: SHAP values for the random masking task.

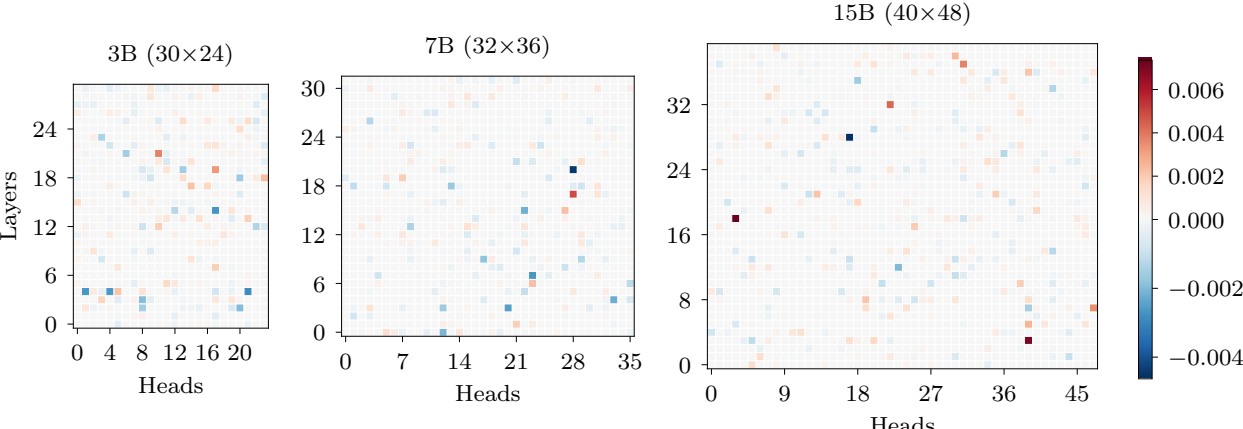

Figure 17: SHAP values for the string literals task.

# B    Intervention all tasks

In this appendix we include all plots for the intervention on all tasks. These are the same plots as the selected plots in Figure 7 (b) and (c), for different tasks.

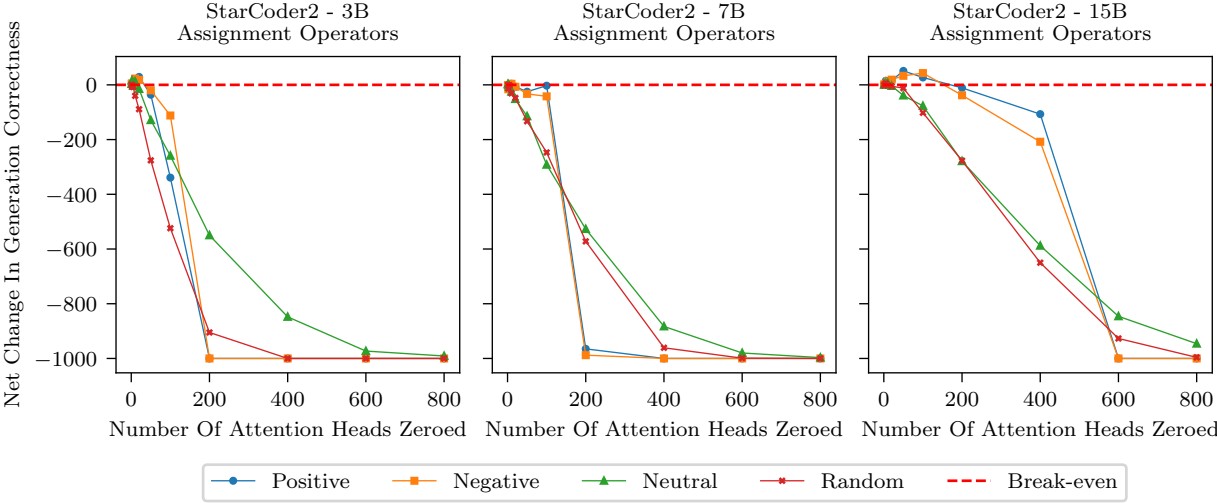

Figure 18: Net change in number of correct generations after intervention for the assignment operators task.

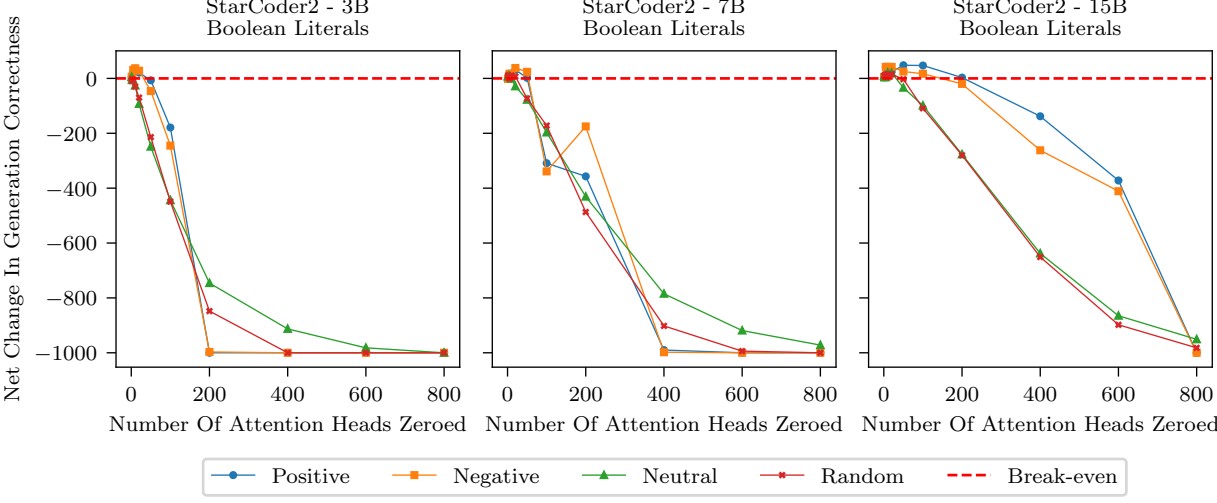

Figure 19: Net change in number of correct generations after intervention for the boolean literals task.

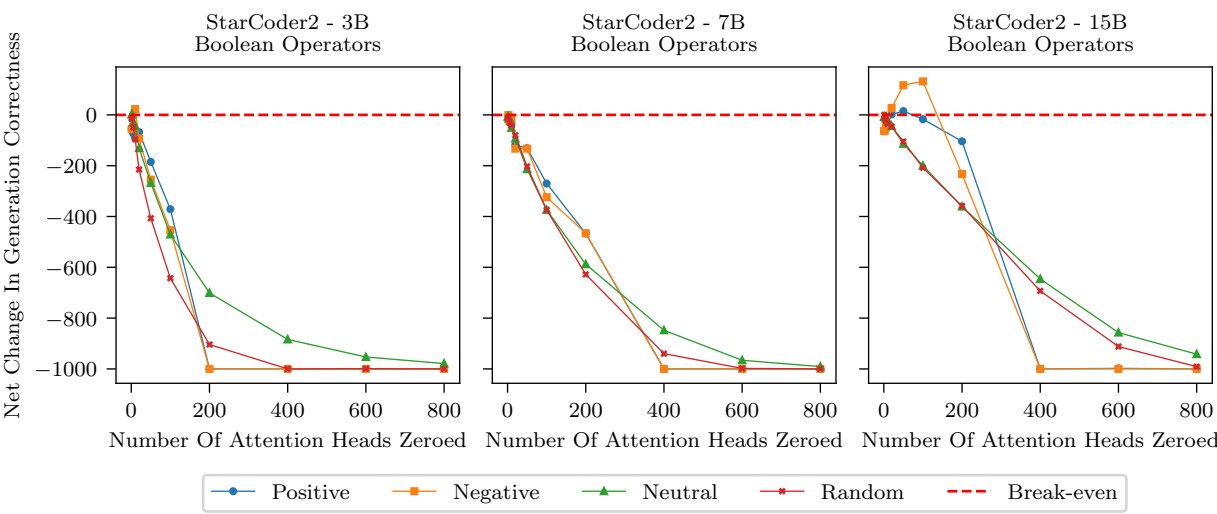

Figure 20: Net change in number of correct generations after intervention for the boolean operators task.

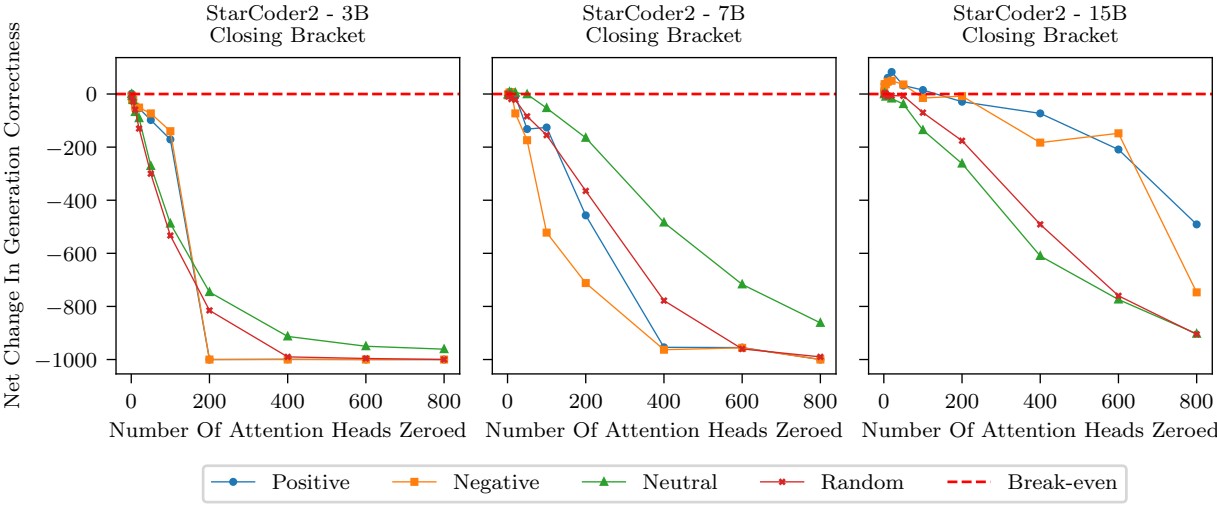

Figure 21: Net change in number of correct generations after intervention for the closing brackets task.

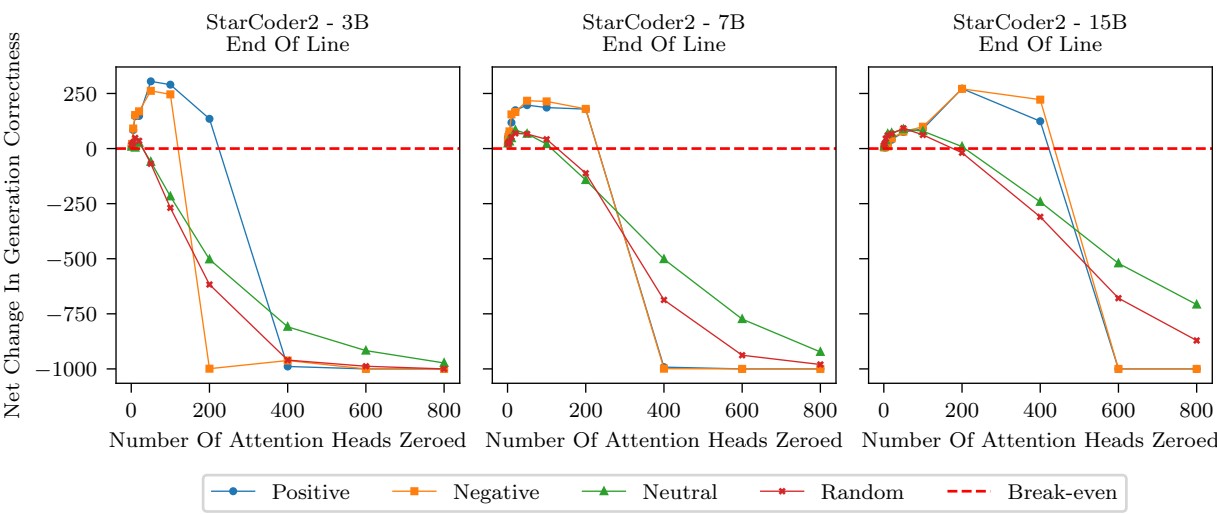

Figure 22: Net change in number of correct generations after intervention for the end-of-line token task.

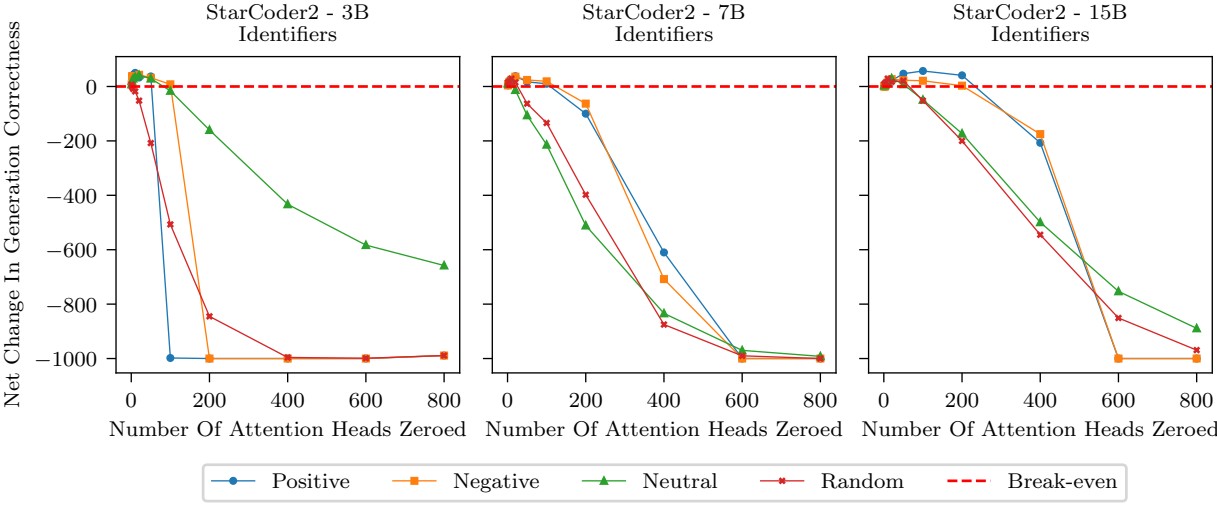

Figure 23: Net change in number of correct generations after intervention for the identifiers task.

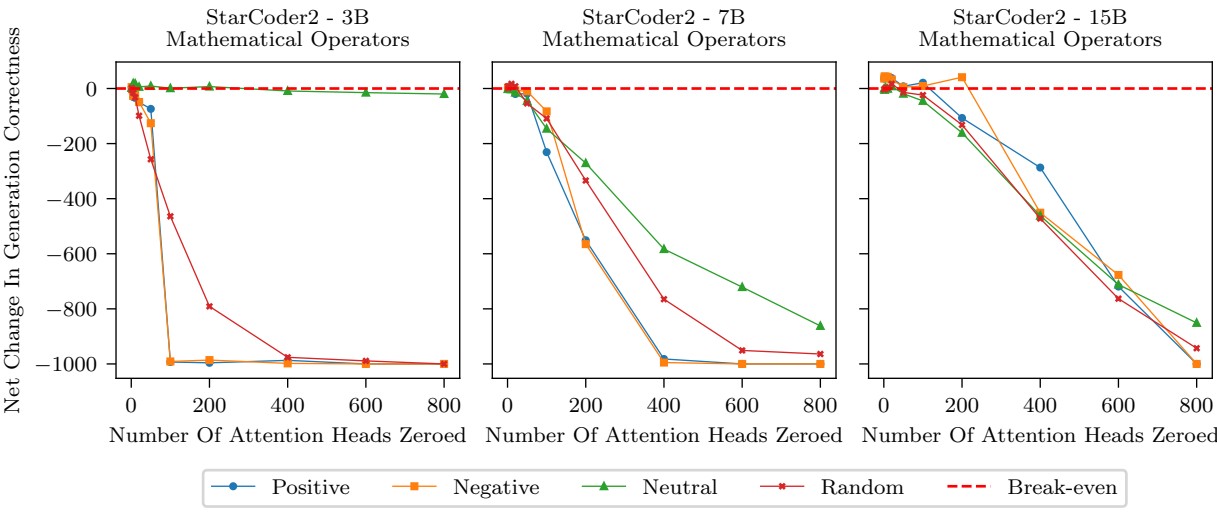

Figure 24: Net change in number of correct generations after intervention for the mathematical operators task.

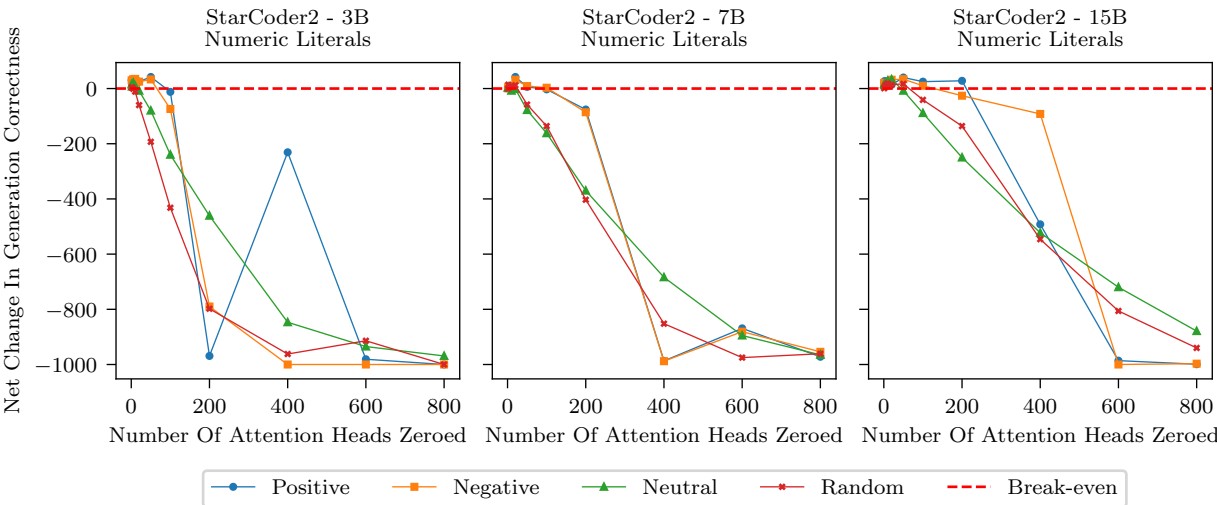

Figure 25: Net change in number of correct generations after intervention for the numeric literals task.

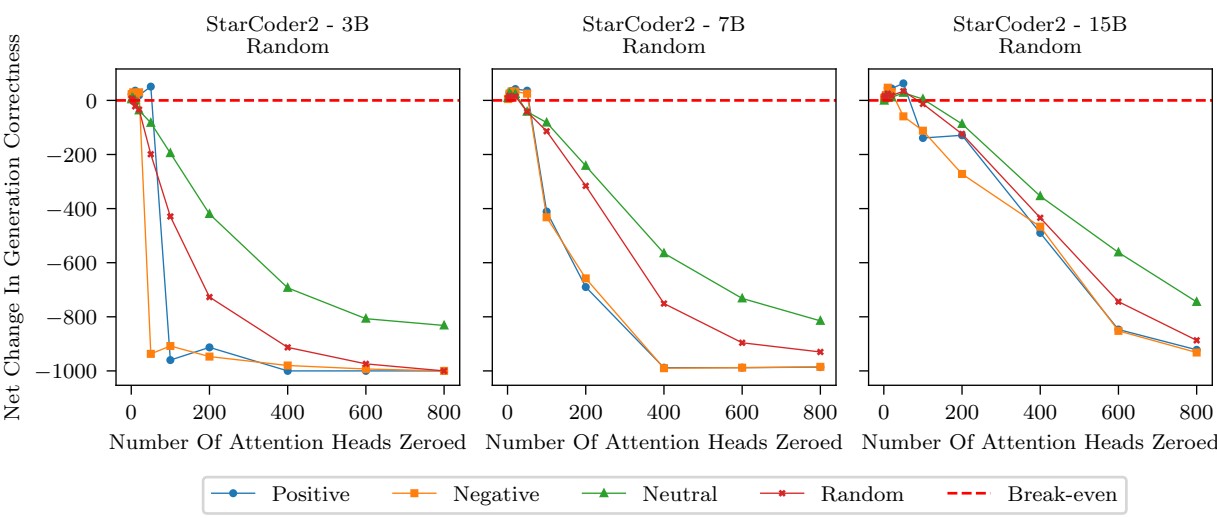

Figure 26: Net change in number of correct generations after intervention for the random masking task.

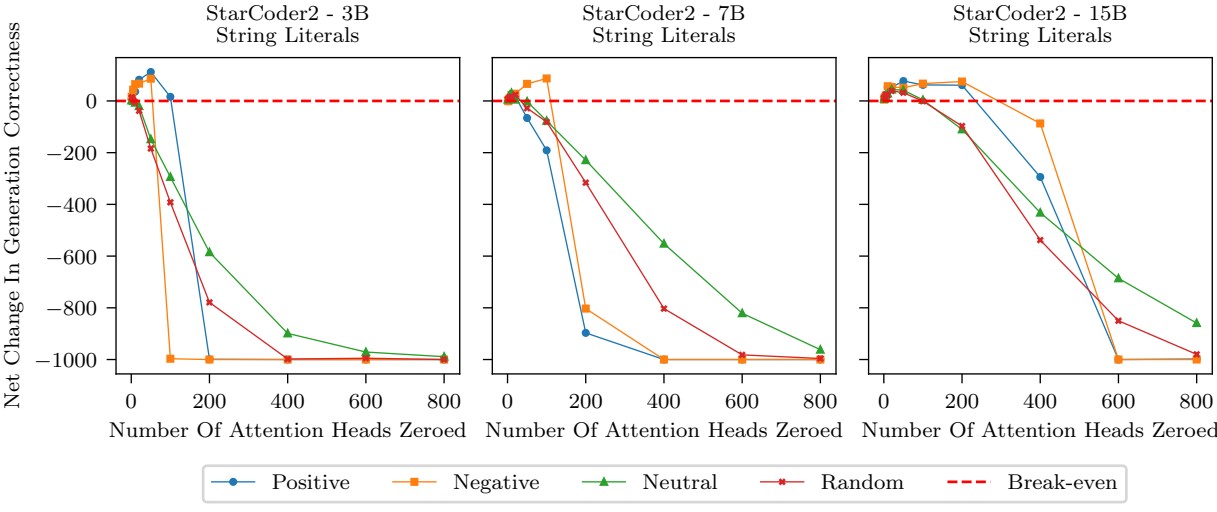

Figure 27: Net change in number of correct generations after intervention for the string literals task.

