# OpenReview forum: "Automated Attention Pattern Discovery at Scale in Large Language Models"
_TMLR — Accepted by TMLR_

### Review · Reviewer_7jFU · 2025-10-15

**Summary Of Contributions:**

This paper introduces the attention pattern - masked autoencoder (AP-MAE), a method for scaling up the interpretability of LLMs. The authors address a key limitation of current mechanistic interpretability techniques: they are often computationally expensive and struggle to generalize to noisy, real world data.
The main idea is to treat the 2D attention patterns from transformer heads as images and use a vision transformer based masked autoencoder (ViT-MAE) to learn a compressed representation of them. The authors apply this method to the StarCoder2 family of code-generation models.

Their main findings are: 1) AP-MAE effectively learns and reconstructs attention patterns, showing that these patterns are a tractable and informative signal. 2) The learned representations generalize well across different model sizes, maybe suggesting the existence of fundamental, shared patterns. 3)  By clustering the learned representations, they identify recurring patterns, some of which correspond to known mechanisms like induction heads, while also discovering novel structures. 4) These pattern representations can be used to predict the correctness of an LLM's next-token generation with up to 70% accuracy, without needing the ground truth.

The authors propose AP-MAE as a scalable "first-pass" tool to identify the most relevant model components for more intensive, fine-grained mechanistic analysis.




Strenghts:
- I think the core idea is very clever and addresses a real problem that is common while working on interpretability projects.
- The intervention experiment (Section 7) is a standout strength. Showing that selectively disabling heads based on the AP-MAE signal can improve model accuracy is a powerful validation of the method. The discovery of the sharp "collapse" threshold when too many heads are removed is an interesting finding in itself, highlighting the model's reliance on a distributed network of components.
- The demonstration that an AP-MAE trained on one model's patterns can effectively reconstruct patterns from another model (Table 2) is a significant result. It suggests the existence of universal attention structures across model scales, making this approach highly efficient.

Weaknesses:
- While the correctness classification accuracy of up to 70% is significantly better than chance, it also indicates that attention patterns alone do not capture the full picture of a model's computation. Similarly, the net accuracy gains from intervention, while positive, are relatively small. This suggests that the signal, while useful, is incomplete.
- The paper could benefit from more discussion on the intuition behind some findings. For example: 1) The ablation study proves that log-scaling is essential, but a brief explanation of why (likely due to the high dynamic range of attention scores where a few tokens receive almost all the weight) would enhance reader understanding. 2) The sudden collapse in performance during intervention is a fascinating result, but the paper doesn't speculate on the mechanism. Is it a cascade failure, or simply the removal of a critical mass of computational capacity?

**Audience:**

Yes

**Audience Explanation:**

The paper addresses a common problem in interpretability and it does that in a clever way.

**Claims And Evidence:**

Yes

**Claims Explanation:**

Especially the intervention results show that this methodology has a good potential.

**Requested Changes:**

- It would be valuable to add a brief discussion contextualizing the performance limits. The authors could acknowledge that attention patterns are just one part of the transformer's computation and hypothesize what other signals (e.g., from MLP layers or the residual stream) might be needed to improve the classification and intervention results further. This would better position their work as a component within a larger interpretability toolkit.

- The paper would be significantly strengthened by a more detailed exploration of the novel patterns discovered. For instance, the authors could select a few examples of the "square-like" patterns and trace the inputs that generate them to hypothesize about their function. This would move the work from pattern discovery towards genuine mechanism discovery.

- A short paragraph in the discussion of the intervention results speculating on the cause of the performance collapse would be very insightful.

---

> ### Author Response · Authors · 2025-12-10
>
> We thank the reviewer for their thoughtful feedback. We combined the weaknesses and requested changes into 4 points, which we will address in the manuscript.
>
> *1 - Attention patterns are only part of the signal.*
>
> We agree with this point. Our motivation for focusing on attention patterns is that relatively little is understood about how different representations are combined within transformer models. In our intervention experiments, we therefore isolate the contribution of the attention pattern by removing only its effect from the computation. Prior work shows that various forms of knowledge also reside in the feedforward layers and other components, so we do not aim to attribute all model behavior to attention alone.
>
> We also note that, in the classification experiments, we are able to detect only those errors that arise from the model failing to perform the intended behavior. It is possible that some errors stem from the model performing correctly according to patterns learned from incorrect data. Such cases would not be captured by our method.
>
> What AP-MAE does provide is a fast and computationally efficient way to highlight regions of a target LLM that are likely to be involved in undesirable behavior. These regions can then be investigated more deeply with higher-resolution tools such as SAEs.
> We will clarify in the paper that AP-MAE is not intended to replace other mechanistic interpretability approaches. Rather, it is designed to complement them by enabling targeted analyses of the specific role played by attention patterns. Together, these methods can contribute to a more complete understanding of how different components of a transformer model participate in the generation of the outputs.
>
> *2 - Scaling of attention patterns*
>
> The reviewer’s intuition regarding the need to scale the attention scores aligns with our own. We applied logarithmic scaling precisely because a small number of tokens receive disproportionately high attention, and this imbalance would otherwise make training on the raw scores difficult. We will include this rationale more explicitly in the manuscript.
>
> *3 - Pattern effect*
>
> Tracing the effects of each attention pattern in detail would require a substantial effort, and a rigorous investigation of each pattern family would merit a dedicated paper of its own. Rather than a limitation, we view this as an opportunity. AP-MAE enables analyses similar to projects such as the OpenAI Microscope, the neuronpedia, or the CNN-filter-DB, where large collections of components are systematically clustered and examined across models. Our method provides the foundation for a similar large-scale exploration of attention patterns, and we see this as a first step toward the more systematic mechanism discovery the reviewer envisions. We will make this forward-looking aspect clearer in the Future Work section.
>
> *4 - Reasons for performance collapse*
>
> Our interpretation of the intervention results is informed by early work in the field, including ACDC. A central criticism of that line of work is that it often identifies a single circuit for a given task, even though it is likely that models rely on multiple backup or secondary circuits that can also yield correct predictions.
>
> Based on this perspective, we speculate that a form of resource competition may be occurring within the model. When a small number of influential heads are removed, alternative circuits can still produce correct predictions. In some cases, removing certain heads may even reduce interference by limiting the number of components attempting to modify the representations for the task. However, once enough of these components are removed, the remaining circuitry becomes insufficient to support correct behavior, and performance degrades.
>
> We will include this interpretation in the results section to provide additional context for the observed intervention effects.

---

### Review · Reviewer_eZ6T · 2025-10-27

**Summary Of Contributions:**

This paper introduces a novel approach for analyzing large language models.
Authors propose to consider attention patterns in a LLM as 2D images, and rely on computer vision analysis methods.
They adapt Vision Transformer Masked Autoencoder (ViT-MAE) to attention pattern analysis, and call the resulting method Attention Pattern Masked Autoencoder (AP-MAE).

The method is very well motivated. In particular, authors contrast the benefit of their approach with limitation of mechanistic interpretability.

For experiments, authors focus on StarCoder2 family of LLMs and the Java language. Data used for experimentation was not used to train the LLMs. They report several metrics, showing that a learned intepretability model AP-MAE generalizes across models (table 2) or allows to intervene on the model (section 7).

**Strenghts:**

- the motivations are clear, and the benefit of such approach compared to previous work are clearly explained by the authors
- although the approach requires to train an auxiliary model, the AP-MAE can generalize across models of different size because it focuses on attention patterns only.

**Weaknesses:**

- Although the motivations are clear, experimental results are not strong enough to support them. Importantly, there is no clear comparison between interpretability methods (see explanation below)
- the authors claims their approach allows to improve LLMs interpretability, but it is unclear what did we learn from this "interpretability results" in experiments
- some experimental details lack clarity, it would be difficult to reproduce experiments from the paper only.

**Audience:**

Yes

**Audience Explanation:**

Interpretability is of interest in deep learning models, especially for LLMs.

**Broader Impact Concerns:**

No concern.

**Claims And Evidence:**

No

**Claims Explanation:**

(1) A major limitation of this work is that experiments focus only on a single family of models and a single domain (i.e. a single programming language). Although evaluating several LLM family in several settings would be too costly, I think that evaluation of another family *OR* another domain/another programming language is required to support the claim.


(2) Moreover, it is unclear to me what is the main claim, and why the proposed method allows to interpret LLM decisions with real-world noisy data (c.f. abstract)? The authors motivate their work by interpretability, but I fail to understand what insight about LLMs prediction is gained with the proposed method. For example, in the intervention section, the authors wrote "suppressing selected heads can improve accuracy in specific tasks, excessive removals change all correct predictions to incorrect", but does this help identify patterns in input that lead to prediction? How does this help us interpret LLMs predictions? Experiments focuses on showing that the approach optimize well some metrics, but I do not see actual interpretability results, in the sense novel insight about how LLMs build predictions for a given task.


(3) Experiments are isolated: authors evaluate their approach, but do not compare with mechanistic interpretability methods. Therefore, it is difficult to agree that AP-MAE is somewhat better than mechanistic interpretability. The paper lacks a convincing experiment of this claim (e.g. "discovered circuits often fail to generalize across tasks, domains, or models" on mechanistic interpretability in the introduction => I don't see clear comparison between AP-MAE and mech. interpret. that shows how AP-MAE is superior w.r.t. this claim, as they do not show comparable experiments including mech. interpret. and AP-MAE).

**Requested Changes:**

Critical recommendation :

- currently, claims are not supported by experiments
- section 5.1: how were these tasks selected? why did the authors chose this subset of tasks? The "tasks" paragraph is very vague on the subject.
- section 5.3 seems contradictory with results in table 2. Authors should give more analysis on this: how can the method both transfer between model sizes while patterns are different? Although I understand it is not contradictory, it is quite surprising.
- section 6.2 is quite hard to follow for readers no familiar with shape values. What is better/what is worse? The current writing is merely on exposition of fact that does not help the reader understand the implication of these experimental results.

Non critical recommendation:

- Figure 1: it would be easier to compare figures if the order was (b), (c), (a), and the caption should help the reader understand what they should look at, what are differences, etc.
- Table 2: caption should give more detail on what is compared in this table. The caption is not stand-alone.
- Figure 2: how where the "noise patterns" generated? + add more information in the caption about what is important in this figure, what should the reader look at.

---

> ### Author Response · Authors · 2025-12-10
> **Explanation about concerns**
>
> Thank you for the constructive feedback. We addressed the concerns below.
>
> *Concern 1 - single model family and single language*
>
> We specifically focus on StarCoder2 (SC2) as the training data is known, and our evaluation corpus (The Heap) is deduplicated against it. This allows us to control for memorization, which is important as it is unknown if memorization impacts attention patterns. We will clarify this in Section 3 and the introduction.
>
> To address the concerns, we extend our experiments by running the AP-MAE pipeline on C++ and Java. The cross-evaluation results show that AP-MAE trained on one language transfers to the other with similar reconstruction loss, and that a multilingual AP-MAE (trained jointly on Java and C++) achieves the lowest reconstruction loss.
>
> **Table 1. Reconstruction loss for patterns generated by Starcoder2 3B for mono and multilingual AP-MAE models: rows = attention patterns trained on, columns = attention patterns evaluated on**
>
> |                     |   Java      |   C++       |
> |---------------------|-------------|-------------|
> | Java Monolingual    | 5.8 ± 0.9   | 5.9 ± 0.9   |
> | C++ Monolingual     | 6.2 ± 0.9   | 6.2 ± 0.9   |
> | Multilingual        | 5.9 ± 0.9   | 5.9 ± 0.9   |
>
> **Table 2. Reconstruction loss for patterns generated by Starcoder2 7B for mono and multilingual AP-MAE models: rows = attention patterns trained on, columns = attention patterns evaluated on.**
>
> |                     |   Java      |   C++       |
> |---------------------|-------------|-------------|
> | Java Monolingual    | 6.6 ± 0.9   | 6.6 ± 0.9   |
> | C++ Monolingual     | 6.7 ± 0.9   | 6.6 ± 0.9   |
> | Multilingual        | 6.4 ± 0.9   | 6.5 ± 0.9   |
>
> **Table 3. Reconstruction loss for patterns generated by Starcoder2 15B for mono and multilingual AP-MAE models: rows = attention patterns trained on, columns = attention patterns evaluated on.**
>
> |                     |   Java      |   C++       |
> |---------------------|-------------|-------------|
> | Java Monolingual    | 7.2 ± 0.9   | 7.2 ± 0.9   |
> | C++ Monolingual     | 7.8 ± 1.0   | 7.7 ± 1.0   |
> | Multilingual        | 7.0 ± 0.9   | 7.0 ± 0.9   |
>
> We also repeated the cross-model evaluation (Table 2) using multilingual AP-MAE models trained on both Java and C++ attention patterns. Across all train/eval combinations, the reconstruction losses remain comparable.
>
> **Table 4. Cross-model generalization with multilingual AP-MAE training. Rows = model size trained on, columns = model size evaluated on.**
>
>
> | Trained on | 3B         | 7B         | 15B         |
> |------------|------------|------------|-------------|
> | **3B**     | 5.9 ± 0.9  | 6.9 ± 0.7  | 8.2 ± 0.9   |
> | **7B**     | 6.4 ± 0.7  | 6.4 ± 0.9  | 8.0 ± 0.9   |
> | **15B**    | 7.4 ± 1.0  | 8.8 ± 1.4  | 7.0 ± 0.9   |
>
> We will add these results to the paper, together with running the classification experiments in both languages. The losses are overall lower as we have found better hyper-parameters since the previous run.
>
> *Concern 2 - Interpretability*
>
> We agree that our approach does not provide the low-level explanations. Our contribution instead offers global interpretability across predictions. We show that AP-MAE can identify heads that cause incorrect generations, and removing these heads leads the model to produce correct outputs. This suggests an internal competition within the model, where components interfere with others during generation. Which improves our understanding of the locality of behaviors. It also shows that attention patterns contain meaningful information, and the global findings can be used to steer fine-grained methods.
>
> *Concern 3 - Comparison to other approaches*
>
> We do not compare our approach to other mechanistic interpretability methods as; these methods are not directly comparable to AP-MAE, and they are prohibitively expensive in the setting we study.
> First, AP-MAE focuses specifically on how representations are mixed, which has not been studied at scale. AP-MAE and representation-based approaches are complementary, each focusing on a separate part of the LLM generation process, and they should cooperate not compete with each other.
> Second, representation based approaches do not generalize well across datasets, inputs, or components of a model [1,2]. This means that a new SAE would need to be trained per head. If we use 3 hours per SAE [3], it takes 3456 hours to train an SAE for each head in SC2 7B.
> Other representation-based methods like replacement networks are more computationally demanding. Training these requires 3,844 H100 GPU hours for a 9B parameter model [4], while AP-MAE takes 100 A40 GPU hours.
>
> [1] https://www.alignmentforum.org/posts/rtp6n7Z23uJpEH7od/saes-are-highly-dataset-dependent-a-case-study-on-the
>
> [2] https://arxiv.org/pdf/2501.16615
>
> [3] https://www.alignmentforum.org/posts/f9EgfLSurAiqRJySD/open-source-sparse-autoencoders-for-all-residual-stream
>
> [4] https://transformer-circuits.pub/2025/attribution-graphs/methods.html

---

> > ### Author Response · Authors · 2025-12-10
> > **Addressing critical recommendations**
> >
> > *1 - Experimental validity.*
> >
> > The validity should be addressed by our answers for all other points.
> >
> > *2 - Task selection*
> >
> > Our task selection follows prior work that identified specific behaviors in LLMs. We agree that there should be a formal definition of a task, however there currently is not. Addressing this would be an investigation in itself.
> > In our study, the choice of tasks only affects the later experiments. Moreover, our SHAP analyses show that patterns appear in distinct locations per task, supporting that the tasks identified by previous work, and adopted in our study, are appropriate and meaningful.
> > We will add a disclaimer to the paper encouraging a formalization of tasks.
> >
> > *3- Section 5.3 vs Table 2*
> >
> > We are happy to clarify in the paper that both results can hold simultaneously
> > Section 5.3 measures within-model diversity across heads: some heads in a LLM exhibit many distinct patterns, while others have very few.
> > Table 2 evaluates across-model similarity of the overall pattern vocabulary: an AP-MAE trained on one model reconstructs patterns from another model with only a small increase in loss, this is due to repeated motifs across models.
> >
> > *4 - SHAP Values*
> >
> > In the text, we will add explanations about what the differences in SHAP values show about how much a certain pattern appearing in a head influences the correctness prediction of the Catboost classifier. Furthermore, we will add notes on the sparsity of the values. Only a few head/pattern combinations are actually required to come up with a prediction. Finally, we will add an explanation about the significance of the locations of high SHAP values in Figure 6, showing that the theory of computational circuits is backed by our findings. We will also add these explanations to the captions of the figures.
> >
> > **Non-critical recommendations**
> >
> > *1 - Figure order*
> >
> > we will change the order of the figures.
> >
> > *2 - Caption Table 2*
> >
> > We will add more explanation to the caption describing what exactly is being evaluated.
> >
> > *3 - Noise*
> >
> > The noise was created by uniformly sampling the vocabulary of the tokenizer. This creates a random input for the LLM that is exactly as long as the context window after tokenization. We will add this to the paper.

---

### Review · Reviewer_bG3p · 2025-12-01

**Summary Of Contributions:**

**Summary**

This paper introduces a promising and scalable method for leveraging LLM attention patterns using a masked autoencoder. Their method AP-MAE, is a vision-based masked autoencoder for reconstructing the attention patterns for the purpose of learning and extracting an embedding which can be analyzed for attention patterns in transformer LLMs at scale. For analysis, they train AP-MAE on millions of StarCoder2 attention maps, and then cluster the embeddings to find recurring structures across heads and uses those clusters to predict next-token correctness with up to ~70% accuracy. Targeted head removals based on SHAP slightly improve accuracy, while removing too many heads collapses performance.

**Strengths**
- This paper is the scalable first step toward analyzing attention patterns.
- The paper is well-written with clear architectural details.
- Many mechanistic techniques are too expensive, overly input-specific, or don’t generalize while this paper argues that attention patterns, unlike residual features, are structurally aligned across models and therefore more scalable for this task.
- There are several interesting insights from the paper as follows:
    - This paper also showcases that attention maps across many heads and layers contain repeated, interpretable 2D motifs.
    - They were able to train a single MAE encoder can reconstruct masked regions of attention maps and generalize across different model sizes (3B ->15B). This indicates that there is a shared attention pattern vocabulary across a family of LLMs.
    - Using nothing but attention pattern clusters per head, they train a CatBoost classifier that can predict correctness of the model’s next-token prediction. Moreover, correct vs incorrect predictions can be distinguished by very sparse sets of heads.
    - These important heads are task-specific as different tasks (identifiers, literals, operators, syntax) highlight completely different sets of important heads.

**Weaknesses**
Three key points remain unclear in this work.
- They compare AP-MAE trained on raw attention to AP-MAE trained on log attention. However, in principle, you can embed attention maps directly without using an MAE. The paper does not include a direct, explicit comparison against such raw or direct embeddings.
- The paper stresses “noisy real-world settings,” but its evaluation is limited to Java code completion from The Heap since, and with most LLM training sets undisclosed, it’s unclear how representative this setting truly is. Hence, it does not test NLP tasks, multi-step reasoning, or conversational behavior. As a result, its claims about general scalability beyond code remain speculative.
- `"As a final step in the training data selection, we focus exclusively on attention patterns generated when the prediction made by the model was correct…"` What was the reason behind this? Please explain the motivation for this strategy. Because AP-MAE is trained only on correct patterns, incorrect ones would be out-of-distribution to the model at inference. Moreover, it skews the representations toward the positive behaviors and limits the detection of dysfunctional or rare failures.

**Minor revision**:
`"predicts whether a generation will be correct without access to ground truth, with up to 70% accuracy..."`
The ~70% accuracy appears only on the best task (end-of-line), not across tasks. It would be helpful to rephrase the claim to reflect the actual numbers.

**Audience:**

Yes

**Audience Explanation:**

This line of work is important because current mechanistic interpretability tools are too computationally expensive to apply across the enormous space of real-world LLM behaviors. Several benchmarks exist for evaluating LLMs, but far less attention has been given to the interpretability of these models.

**Claims And Evidence:**

Yes

**Claims Explanation:**

The paper is well-written and provides sufficient experiments to support its claims.

**Requested Changes:**

Please read weaknesses.

---

> ### Author Response · Authors · 2025-12-10
> **Response**
>
> Thank you for the valuable comments. Here are our responses and how we will address them.
>
> *1- comparison to direct embeddings (assuming the reviewer meant techniques such as VLAD/PCA)*
>
> We understand the concern. To avoid confusion, we clarify our goal in this work. We do not aim to argue that AP-MAE is the only or optimal way to embed attention patterns, but to show that attention patterns have consistent shapes, and that they are at least partially informative about model behavior. In this work, we show AP-MAE is one concrete instantiation of this idea.
>
> We used a transformer-based approach because before AP-MAE, not much was known about attention patterns outside of a select few heads in a circuit. This means that the patterns could be very complex, leading to the need for sophisticated pattern recognition tools to detect and abstract away from potential noise. Transformer models are known for their robustness to noisy data.
> Importantly, our main claims do not rely on AP-MAE being the optimal encoder. The key results depend on having some embedding that preserves pattern structure, not on the specific architecture of AP-MAE.
> Having seen the consistency of attention patterns with AP-MAE, we fully agree that alternative embedding approaches are worth discussing, and we will revise the manuscript accordingly. In particular:
> - In the Limitations section, we will broaden our discussion from CNNs embedding methods in general (e.g., PCA, VLAD, CNNs), and clarify that AP-MAE is a design choice rather than a core contribution.
> - We will add a short paragraph in Section 5 (Pattern Mining) explaining why the reconstruction objective is useful for validating that the embeddings indeed capture global motifs, and how a direct-embedding baseline could be plugged into the same clustering + classification pipeline.
>
> *2- On the scope of “noisy real-world settings”*
>
> We appreciate this clarification request. In this paper, “real-world” contrasts with carefully hand-crafted prompts and templates, not as a claim about all possible natural-language or multi-step reasoning tasks. Our experiments are conducted on Java code completion tasks, mined automatically from The Heap. These inputs preserve the noise and irrelevant context that occurs in practice (e.g., long methods where large portions of the surrounding code do not affect a boolean value prediction), in contrast to many mechanistic interpretability works that rely on short, template-like inputs. Our contributions are therefore about scaling attention-pattern analysis in this realistic code-completion setting, not about demonstrating coverage over all domains or tasks.
> To avoid ambiguity, we will revise the Abstract and contribution bullets to say “noisy, real-world code completion data from The Heap” instead of “noisy real-world settings”. We will also add an explicit sentence to the Limitations stating that our empirical claims are restricted to the studied programming languages and the StarCoder2 family, and that extending AP-MAE to natural-language, multi-step reasoning, and conversational tasks is an important direction for future work.
>
> *3- Filtering of incorrect completions during training*
>
> The decision to filter out incorrect completions is based on four factors.
> First, an incorrect completion may create an underdeveloped attention pattern. If these patterns emerge as artifacts of the LLM training process, then for certain completions the model may fail to form a meaningful attention structure. Including such cases in training could therefore introduce noise.
> Second, previous studies have shown that some specific tokens, often referred to as adversarial or glitch tokens, can cause models to produce incorrect predictions. We hypothesize that these tokens also distort attention patterns in ways we do not want the model to learn from. Because our training data is based on real-world inputs, we cannot guarantee that such tokens are never present, we can limit their effect by selecting for correct predictions.
> Finally, as we are using the embeddings for a classification task, where we cluster with HDBSCAN, the OOD patterns that could be generated with an incorrect prediction will either be rare and classified as outliers (seen as one category for catboost), or very common and form a cluster themselves.
> We agree that a dedicated analysis of failure-mode patterns is an interesting direction for future work. Our current objective, however, is to first establish a robust representation of common attention motifs; once that foundation is in place, AP-MAE can be reused as a tool to study dysfunctional or rare behaviors more systematically
>
> *4-On the “up to 70% accuracy” wording*
>
> Thanks for pointing this out. We will change the text to:
>
> “predicts whether a generation will be correct without access to ground truth, with accuracies ranging from ≈55% to 70% depending on the task, and reaching 70% on the best-performing task, End-of-line token prediction.”

---

### Decision · Action_Editor_Z8xb · 2026-01-09

**Recommendation:** Accept with minor revision

**Additional Comments:**

I agree with the reviewers that there is valuable insight in this work that the community will find useful. As the submitted version had some ambiguity about the scope of the claims, and the interpretation of some of the results, I request that the changes promised in the response to reviews be incorporated in the final manuscript. I also additionally highlight some more requests coming from reviewers and from my own reading of the discussion:

- Discuss how to expand the analysis beyond just code completion to more diverse domains. Mainly outline directions that would be important for the community to build upon this line of work.
- Clarify how to use the presented approach for interpretability.
- Include a justification of the choice to not compare with other methods.
- Include a justification of filtering out incorrect completions during training.

Promised:
- Broaden limitation discussion from CNNs; clarify AP-MAE design choice.
- Add paragraph in S5 about reconstruction objective.
- Add context to "noisy real-world data" claim in Abstract, and overall about scope of experiments/data
- Change the "up to 70%" wording.
- Add disclaimer about "formalization of tasks"
- Fig 1: reorder subfigures as b, c, a; consider improving caption.
- Table 2 clarify caption
- Figure 2 (or narrative text): clarify noise.
- Justify log-scaling of attention scores.
- Include discussion of performance collapse.

I will check the revision. To make this easier on me, I would appreciate if you could somehow highlight the changes. One suggestion is to first submit a revision where the changes are highlighted in red, and then another revision with the highlighting removed (I have access to all revisions in openreview). Another option is to post the (searchable) new text added for each request in a forum comment visible to me.

**Audience:**

Yes

**Audience Explanation:**

It is a fresh perspective on data-driven analysis of model internals that I find (and agree with reviewers that it is) relevant to many in the community

**Claims And Evidence:**

Yes

**Claims Explanation:**

Following clarifications in the author responses, the reviewers do find that the claims are supported, if phrased somewhat more clearly.